# Thermal Analysis of a Magnetic Brake Using Infrared Techniques and 3D Cell Method with a New Convective Constitutive Matrix

**DOI:** 10.3390/s19092028

**Published:** 2019-04-30

**Authors:** José Miguel Monzón-Verona, Pablo Ignacio González-Domínguez, Santiago García-Alonso, Francisco Jorge Santana-Martín, Juan Francisco Cárdenes-Martín

**Affiliations:** 1Electrical Engineering Department, University of Las Palmas de Gran Canaria, 35017 Las Palmas de Gran Canaria, Spain; josemiguel.monzon@ulpgc.es (J.M.M.-V.); fjorge.sanmar@gmail.com (F.J.S.-M.); 2Institute for Applied Microelectronics, University of Las Palmas de Gran Canaria, 35017 Las Palmas de Gran Canaria, Spain; 3Department of Electronic Engineering and Automatics (DIEA), University of Las Palmas de Gran Canaria, 35017 Las Palmas de Gran Canaria, Spain; santiago.garciaalonso@ulpgc.es; 4Mechanical Engineering Department, University of Las Palmas de Gran Canaria, 35017 Las Palmas de Gran Canaria, Spain; juanfrancisco.cardenes@ulpgc.es

**Keywords:** Cell Method, infrared sensors, magnetic brake, thermography

## Abstract

In this work we analyse the temperature distribution in a conductor disk in transitory regime. The disk is in motion in a stationary magnetic field generated by a permanent magnet and so, the electric currents induced inside it generate heat. The system acts as a magnetic brake and is analysed using infrared sensor techniques. In addition, for the simulation and analysis of the magnetic brake, a new thermal convective matrix for the 3D Cell Method (CM) is proposed. The results of the simulation have been verified by comparing the numerical results with those obtained by the Finite Element Method (FEM) and with experimental data obtained by infrared technology. The difference between the experimental results obtained by infrared sensors and those obtained in the simulations is less than 0.0459%.

## 1. Introduction

Thermal and electromagnetic analysis are strongly linked and they are applied in the design of all electrical machines [1,2]. In particular, transient thermal analysis is applied in magnetic brakes [3]. The use of magnetic brakes has obvious advantages over brakes based on mechanical friction. The latter have the risks of hydraulic fluid loss and contamination of the fluid by cooling water, with a consequent loss of braking power, among others [4]. The use of permanent magnets in linear magnetic brakes is explained in detail in [5]. In [4] the results obtained are compared with an analytical equation and with 3D FEM. In [6] the non-linear transient analysis of magnetic brakes used by high-speed trains is studied.

In most of the works consulted in the bibliography dealing with the thermal analysis of magnetic brakes, approximate analytical equations are used, or numerical methods based on the differential formulation, such as the FEM, are employed.

In the present work we propose the Finite Formulation—FF [7], and the CM [8,9] as an associated numerical method to analyse this type of devices. In this methodology we work with global magnitudes associated to space oriented elements such as volumes, surfaces, lines and points of the discretized space, as well as to temporal elements, instead of field magnitudes associated to independent variables –spatial and temporal coordinates [10].

In addition, the equations of constitutive type—equations of the medium—are clearly differentiated from the topological type—equations of balance. In FF, the physical laws that govern the electromagnetic equations and the thermal laws of heat transfer associated with magnetic brakes, are expressed in their integral form. In this way, the final system of equations is posed directly, without the need to discretize the equivalent differential equations [11].

The thermal analysis of the magnetic brake using this methodology greatly facilitates the conditions of contour and continuity, when working with global magnitudes and directly raise the system of equations without the need to discretize the differential equations.

In the present work we have formulated a new constitutive matrix that relates the flows of convective heat power—magnitude type source—due to the movement of the disk, with the magnitudes of configuration—temperatures—using CM. The magnitudes of configuration are associated to the nodes of a primal mesh made up of tetrahedra, and the source type magnitudes are associated to the surfaces of a dual mesh—control volume—obtained in a barycentric division of the primal mesh. This matrix is applied to an equation of energy balance on a magnetic brake, which consists of a copper disk that rotates with an angular velocity *w_r_*, in a magnetic field produced by a permanent magnet.

The new thermal convective matrix formulated with CM in 3D has been verified by contrasting the numerical results with those obtained by FEM. The difference between the experimental results obtained by infrared sensors and those obtained in the simulations is less than 0.0459%.

To obtain the experimental validation of the new convective matrix it is necessary to measure the temperature of a rotating disk in a magnetic field without physical contact between thermometer and disk [12]. Besides, it is desired that the sensor have a low thermal inertia and that its response is reasonably fast, that it provides a high resolution in its measure and that it is easy to use.

Therefore, in the present work the use of infrared thermometers is required since these thermometers can measure the temperature of an object by detecting the infrared energy emitted by all the materials involved in the experiment.

These thermometers are constituted by a lens that focuses infrared energy on a detector that converts this energy into an electrical signal that once processed can be expressed in units of temperature. This configuration facilitates the measurement of the temperature of an object at a distance without having contact with it. It is useful to measure the temperature where another type of sensor cannot be used.

Infrared thermometers typical applications are, to measure temperatures of objects that are in movement, or surrounded by an electromagnetic field, or in vacuum conditions. Due to the lack of mechanical inertia caused by their electrical nature, they are used in applications where a fast response is desired.

We have used two different infrared sensors: a punctual thermal sensor and a camera, to verify the new convective thermal matrix. One takes measures in a point and the other takes measures in matrix array. The measures of both methods have been compared with FEM and CM.

This work has been divided into the following sections: Section 2 explains in detail the methodology for obtaining the new constitutive matrix in the CM, formulating the corresponding convective term. In Section 3 the fundaments of the measurement with infrared sensors are exposed. In Section 4 the numerical and experimental validation of the results of the thermal transient regime is carried out. Finally, Section 5 presents the conclusions.

## 2. The Constitutive Convective Matrix in the CM

In this section, after studying the electromagnetic equations in the CM, the thermal formulation is proposed in the time domain for the CM and the new thermal convective constitutive matrix is formulated. Finally, thermal problem boundary conditions are applied. The new convective constitutive thermal matrix deduced in this section will be experimentally validated through the experimental data obtained in Section 4.

### 2.1. Electromagnetic Equations in the CM

The thermal equations developed in this work are applied to a magnetic brake. This basically consists of a copper disk and a permanent magnet. The magnet is located on the disk, at a distance *d*. The disk is mechanically coupled to a DC motor through six screws. It rotates it at an angular velocity *w_r_*, as Figure 1 shows. To solve the thermal equations using the CM, it is necessary to know the sources of heat in the dual volume, see Figure 2. 

These are obtained by solving the electromagnetic Equations (1) that are explained in detail in [13]:(1)$$[\begin{array}{cc} C^{t}M_{v}C-M_{\sigma }V_{Lo}C+jWM_{\sigma } & M_{\sigma }G\\ -G^{t}M_{\sigma }jW+G^{t}M_{\sigma }V_{Lo}C & -G^{t}M_{\sigma }G \end{array}]\left[\begin{array}{c} a\\ \phi \end{array}\right]=\left[\begin{array}{c} C^{t}F_{e}\\ 0 \end{array}\right]$$

The system of equations (1) is obtained for each tetrahedron. The assembly is done by passing a loop through all the tetrahedra of the domain, where the degrees of freedom *a* and ϕ are the magnetic potential and electric potential, respectively. In this system, the matrix $M_{\sigma }=\frac{\sigma^{e}}{v^{e}}{\tilde{S}}_{i}{\tilde{S}}_{j}\;i,j=1:6$, is called the electrical conductivity constitutive matrix. This matrix is a function of the conductivity of each tetrahedron, of the volume of each tetrahedron and of the scalar products of the surface vectors of the dual planes to the edges *e_i_* and *e_j_* in the tetrahedron [14]. $M_{v}$ is the reluctivity matrix and $F_{e}$ is the vector of coercive magnetomotive forces obtained from the manufacturer of the permanent magnet.

The matrix C shown in (2) gives us the incidences between the faces and the edges in the reference tetrahedron—it represents the discrete rotational operator—, see Figure 3b, and G is the incidence matrix between the edges and the nodes of the reference tetrahedron –represents the discrete gradient–, see Figure 3b:(2)$$C=\left[\begin{array}{cccccc} 0 & 1 & 0 & 0 & 0 & 0\\ 0 & 0 & -1 & 1 & 1 & -1\\ 1 & 0 & 0 & -1 & -1 & 0\\ -1 & -1 & 1 & 0 & 0 & 0 \end{array}\right],G=\left[\begin{array}{cccc} -1 & 1 & 0 & 0\\ 0 & -1 & 1 & 0\\ -1 & 0 & 1 & 0\\ -1 & 0 & 0 & 1\\ 0 & -1 & 0 & 1\\ 0 & 0 & -1 & 1 \end{array}\right].$$

$V_{Lo}$ is calculated by:(3)$$V_{Lo}=\frac{1}{3V_{t}}\left[\begin{array}{cccc} {\vec{V}}_{e}\times {\vec{e}}_{4}\cdot {\vec{e}}_{1} & {\vec{V}}_{e}\times {\vec{e}}_{5}\cdot {\vec{e}}_{1} & {\vec{V}}_{e}\times {\vec{e}}_{6}\cdot {\vec{e}}_{1} & 0\\ {\vec{V}}_{e}\times {\vec{e}}_{4}\cdot {\vec{e}}_{2} & {\vec{V}}_{e}\times {\vec{e}}_{5}\cdot {\vec{e}}_{2} & {\vec{V}}_{e}\times {\vec{e}}_{6}\cdot {\vec{e}}_{2} & 0\\ {\vec{V}}_{e}\times {\vec{e}}_{4}\cdot {\vec{e}}_{3} & {\vec{V}}_{e}\times {\vec{e}}_{5}\cdot {\vec{e}}_{3} & {\vec{V}}_{e}\times {\vec{e}}_{6}\cdot {\vec{e}}_{3} & 0\\ {\vec{V}}_{e}\times {\vec{e}}_{4}\cdot {\vec{e}}_{4} & {\vec{V}}_{e}\times {\vec{e}}_{5}\cdot {\vec{e}}_{4} & {\vec{V}}_{e}\times {\vec{e}}_{6}\cdot {\vec{e}}_{4} & 0\\ {\vec{V}}_{e}\times {\vec{e}}_{4}\cdot {\vec{e}}_{5} & {\vec{V}}_{e}\times {\vec{e}}_{5}\cdot {\vec{e}}_{5} & {\vec{V}}_{e}\times {\vec{e}}_{6}\cdot {\vec{e}}_{5} & 0\\ {\vec{V}}_{e}\times {\vec{e}}_{4}\cdot {\vec{e}}_{6} & {\vec{V}}_{e}\times {\vec{e}}_{5}\cdot {\vec{e}}_{6} & {\vec{V}}_{e}\times {\vec{e}}_{6}\cdot {\vec{e}}_{6} & 0 \end{array}\right],$$where the velocity of each point of the disk, ${\vec{V}}_{e}$ is given by:(4)$${\vec{V}}_{e}={\vec{W}}_{r}\times \vec{r}=\left|\begin{array}{c} \begin{array}{ccc} \vec{i} & \vec{j} & \vec{k} \end{array}\\ \begin{array}{ccc} 0 & w_{r} & 0 \end{array}\\ \begin{array}{ccc} r_{x} & r_{y} & r_{z} \end{array} \end{array}\right|=\left(w_{r}r_{x},0,-r_{x}w_{r}\right),$$where ${\vec{W}}_{r}$ the angular velocity of the disk, $\vec{r}$ is the radius vector with respect to its centre, ${\vec{e}}_{i}\;i=1:6$ are the edges associated with the tetrahedron of the primal mesh, $V_{t}$ is the volume of each tetrahedron, see Figure 3b. 

### 2.2. Thermal Formulation in the Time Domain in the CM

Applying the energy balance equation to the dual volume of control [15], see Figure 3a, we obtain the following equation:(5)$$M_{\rho C_{p}}\frac{dT}{dt}+\tilde{D}\left(-M_{\lambda }GT\right)=\tilde{W},$$where $\tilde{W}$ is obtained once the system of equations (1) has been solved, $\tilde{D}$ represents the discrete divergence associated with the dual volume, being $\tilde{D}=-G^{t}$, $M_{\rho C_{p}}$ is the constitutive matrix in heat transmission in transitory state and $M_{\lambda }$ is the thermal conductivity constitutive matrix. In this way, we obtain the heat sources in the disk, which are calculated as follows:(6)$$\tilde{W}=\frac{1}{\sigma }\vec{J}\cdot \vec{J}d\tilde{V},$$where $\vec{J}$ is calculated, by the following equation:(7)$$\vec{J}=\sigma \left(-\vec{grad}\phi +\vec{v}\times \vec{B}\right),$$ being *σ* the electrical conductivity, ϕ the electric scalar potential, $\vec{v}\;$ the velocity of each point and $\vec{B}$ is the magnetic induction. See [13].

In Equation (6) $\tilde{W}$ are the heat sources associated with the dual volumes $\tilde{V}$, four for each tetrahedron. In Equation (5), the term $\left(-M_{\lambda }GT\right)$ represents the Fourier law of heat transfer by conduction in the CM, associated to the four dual surfaces (see Figure 3b). The matrix $M_{\lambda }$ is the thermal conductivity matrix [16], *T* are the temperatures associated to the primal nodes. That is to say, ${\tilde{w}}_{\lambda }=-M_{\lambda }GT$.

### 2.3. New Thermal Convective Constitutive Matrix

The disk moves with respect to the control volume. Therefore, we must add a flow of heat of the type:(8)$${\tilde{w}}_{\rho C_{p}v}=M_{\rho C_{p}v}T.$$

In this way, the total heat power flow associated with the dual surfaces will be the purely conductive plus the purely convective due to the movement of the disk mass, as expressed below:(9)$${\tilde{w}}_{t}={\tilde{w}}_{\rho C_{p}v}+{\tilde{w}}_{\lambda }.$$

Our contribution is to calculate a new convective constitutive matrix $M_{\rho C_{p}v}$ that allows us to obtain Equation (8). This addition takes into account the heat flow due to the rotation movement of the disk. In this way, Equation (5) would be transformed into the following equation:(10)$$M_{\rho C_{p}}\frac{dT}{dt}+\tilde{D}\left(-M_{\lambda }GT+M_{\rho C_{p}v}T\right)=\tilde{W}.$$

To obtain Equation (8), we start from the field equation of heat power flow for any point within the tetrahedron, which is expressed as follows:(11)$${\vec{w}}_{\rho C_{p}v}=\rho C_{p}{\vec{V}}_{e}T\left(\lambda_{1},\lambda_{2},\lambda_{3},\lambda_{4}\right),$$where ${\vec{V}}_{e}$ is the barycenter velocity of each tetrahedron and $T\left(\lambda_{1},\lambda_{2},\lambda_{3},\lambda_{4}\right)$ is the temperature at any point within the tetrahedron as a function of the barycentric coordinates of the tetrahedron, where these are $\lambda_{i}\in \left[0,1\right]\;i=1:4$, such that $\lambda_{1}+\lambda_{2}+\lambda_{3}+\lambda_{4}=1$. $T\left(\lambda_{1},\lambda_{2},\lambda_{3},\lambda_{4}\right)$ is calculated as shown below:(12)$$T\left(\lambda_{1},\lambda_{2},\lambda_{3},\lambda_{4}\right)=\lambda_{1}T_{1}+\lambda_{2}T_{2}+\lambda_{3}T_{3}+\lambda_{4}T_{4},$$where $T_{i}\;i=1:4$ are the unknowns associated with the tetrahedron nodes.

To obtain the constitutive matrix we calculate the heat flux ${\tilde{w}}_{\rho C_{p}v}^{3}$ in the dual plane ${\tilde{S}}_{3}$ (see Figure 3a,b). In the rest of the dual planes is done in the same way. Then, we must calculate the following integral:(13)$${\tilde{w}}_{\rho C_{p}v}^{3}=\overset{}{\underset{{\tilde{S}}_{3}}{\int }}\rho C_{p}{\vec{V}}_{e}T\left(\lambda_{1},\lambda_{2},\lambda_{3},\lambda_{4}\right)\cdot {\vec{n}}_{3}ds.$$

Substituting (12) in (13), we obtain:(14)$${\tilde{w}}_{\rho C_{p}v}^{3}=\overset{}{\underset{{\tilde{S}}_{3}}{\int }}\rho C_{p}{\vec{V}}_{e}\left(\lambda_{1}T_{1}+\lambda_{2}T_{2}+\lambda_{3}T_{3}+\lambda_{4}T_{4}\right)\cdot {\vec{n}}_{3}ds,$$where ${\vec{n}}_{3}$ is a vector perpendicular the dual plane ${\tilde{S}}_{3}$. To calculate this integral, we use the following exact integral:(15)$$\overset{}{\underset{S_{ij}}{\int }}\lambda_{\delta }ds=\{\begin{array}{l} \frac{5}{432}\left|S_{ij}\right|\;v=i\;or\;j\\ \frac{13}{432}\left|S_{ij}\right|\;v\neq i\;or\;j\;. \end{array}$$

The correspondence of notations between those found in [17] and those shown in Figure 3b will be as follows: ${\tilde{S}}_{1}\Leftrightarrow {\tilde{S}}_{34}$, ${\tilde{S}}_{2}\Leftrightarrow {\tilde{S}}_{14}$, ${\tilde{S}}_{3}\Leftrightarrow {\tilde{S}}_{42}$, ${\tilde{S}}_{4}\Leftrightarrow {\tilde{S}}_{23}$, ${\tilde{S}}_{5}\Leftrightarrow {\tilde{S}}_{13},\;{\tilde{S}}_{6}\Leftrightarrow {\tilde{S}}_{12}$. Then the integral (14) is expressed as:(16)$${\tilde{w}}_{\rho C_{p}v}^{3}=\rho C_{p}{\vec{V}}_{e}\cdot {\vec{n}}_{3}\frac{{\tilde{S}}_{3}}{432}\left(13T_{1}+5T_{2}+13T_{3}+5T_{4}\right).$$

Making ${\vec{n}}_{3}{\tilde{S}}_{3}={\vec{\tilde{S}}}_{3}$, then, analogously, it is done with the rest of the dual planes and the constitutive matrix is obtained:(17)$$\left[\begin{array}{c} \begin{array}{c} {\tilde{w}}_{\rho C_{p}v}^{1}\\ {\tilde{w}}_{\rho C_{p}v}^{2}\\ {\tilde{w}}_{\rho C_{p}v}^{3} \end{array}\\ \begin{array}{c} {\tilde{w}}_{\rho C_{p}v}^{4}\\ {\tilde{w}}_{\rho C_{p}v}^{5}\\ {\tilde{w}}_{\rho C_{p}v}^{6} \end{array} \end{array}\right]=\frac{\rho C_{p}{\vec{V}}_{e}}{432}\left[\begin{array}{cccc} 13{\vec{\tilde{S}}}_{1} & 13{\vec{\tilde{S}}}_{1} & 5{\vec{\tilde{S}}}_{1} & 5{\vec{\tilde{S}}}_{1}\\ 5{\vec{\tilde{S}}}_{2} & 13{\vec{\tilde{S}}}_{2} & 13{\vec{\tilde{S}}}_{2} & 5{\vec{\tilde{S}}}_{2}\\ 13{\vec{\tilde{S}}}_{3} & 5{\vec{\tilde{S}}}_{3} & 13{\vec{\tilde{S}}}_{3} & 5{\vec{\tilde{S}}}_{3}\\ 13{\vec{\tilde{S}}}_{4} & 5{\vec{\tilde{S}}}_{4} & 5{\vec{\tilde{S}}}_{4} & 13{\vec{\tilde{S}}}_{4}\\ 5{\vec{\tilde{S}}}_{5} & 13{\vec{\tilde{S}}}_{5} & 5{\vec{\tilde{S}}}_{5} & 13{\vec{\tilde{S}}}_{5}\\ 5{\vec{\tilde{S}}}_{6} & 5{\vec{\tilde{S}}}_{6} & 13{\vec{\tilde{S}}}_{6} & 13{\vec{\tilde{S}}}_{6} \end{array}\right]\left[\begin{array}{c} \begin{array}{c} T_{1}\\ T_{2}\\ T_{3} \end{array}\\ T_{4} \end{array}\right]=M_{\rho C_{p}v}T.$$

### 2.4. Boundary Conditions of the Thermal Problem

The boundary condition of the thermal problem on the surface of the disk in the differential formulation is that indicated below:(18)$$-k\;\vec{grad}\;T\cdot \vec{n}+\rho C_{p}{\vec{V}}_{e}T\left(\lambda_{1},\;\lambda_{2},\;\lambda_{3},\;\lambda_{4}\right)\cdot \vec{n}=h_{eff}\left(T-T_{am}\right)=q_{s},$$where:(19)$$h_{eff}\left(T-T_{am}\right)=h_{t}\left(T-T_{am}\right)+\varepsilon \sigma_{SB}\left(T^{4}-T_{am}^{4}\right),$$where *k* is the thermal conductivity of the copper disk, *T* is the temperature at the disk surface, $\vec{n}$ is a normal vector to the disk surface, $T_{am}$ is the ambient temperature, *h_t_* is the thermal convective heat transfer coefficient between the solid and the air that surrounds the disk. $h_{eff}$ is the effective heat transfer coefficient that takes into account the convective effect—which is the predominant one–, as well as the radiation effect, being *ε* the emissivity factor and *σ_SB_* the Stefan-Boltzmann constant.

In the CM, the corresponding equation is calculated in relation to the dual surfaces that belong to the contour (see Figure 4).

The temperature at any point of the triangle of the contour can be expressed as the interpolation of the temperatures of the nodes of the triangle as a function of its barycentric coordinates:(20)$$T=\lambda_{1}T_{1}+\lambda_{2}T_{2}+\lambda_{3}T_{3}.$$

Substituting Equation (20) in Equation (18), and integrating in each dual surface of the contour triangle, we obtain:(21)$${\tilde{w}}_{conv}^{1}=\overset{}{\underset{{\tilde{S}}_{1}}{\int }}h_{eff}\left(T-T_{am}\right)ds=\overset{}{\underset{{\tilde{S}}_{1}}{\int }}h_{eff}\left(\lambda_{1}T_{1}+\lambda_{2}T_{2}+\lambda_{3}T_{3}-T_{am}\right)ds.$$

Applying the formulas of the exact integration (22), Equation (21) is transformed into (23), see [17]:(22)$$\overset{}{\underset{{\tilde{S}}_{k}}{\int }}\lambda_{\delta }ds=\{\begin{array}{l} \frac{11}{18}\left|{\tilde{S}}_{k}\right|\;\delta =k\\ \frac{7}{36}\left|{\tilde{S}}_{k}\right|\;\delta \neq k, \end{array}$$
(23)$${\tilde{w}}_{conv}^{1}=\frac{h_{eff}\Delta }{3}\left(\frac{11}{18}T_{1}+\frac{7}{36}T_{2}+\frac{7}{36}T_{3}\right)-\frac{h_{eff}\Delta T_{am}}{3},$$ where Δ is the area of each triangle of the contour. Analogously it is done with the other two dual surfaces, leaving the final Equation (24). This equation is equivalent to that obtained in [15] in purely geometric form:
(24)$$\left[\begin{array}{c} {\tilde{w}}_{conv}^{1}\\ {\tilde{w}}_{conv}^{2}\\ {\tilde{w}}_{conv}^{3} \end{array}\right]=\frac{h_{eff}\Delta }{3}\left[\begin{array}{ccc} \begin{array}{c} \frac{11}{18}\\ \frac{7}{36}\\ \frac{7}{36} \end{array} & \begin{array}{c} \frac{7}{36}\\ \frac{11}{18}\\ \frac{7}{36} \end{array} & \begin{array}{c} \frac{7}{36}\\ \frac{7}{36}\\ \frac{11}{18} \end{array} \end{array}\right]\left[\begin{array}{c} T_{1}\\ T_{2}\\ T_{3} \end{array}\right]-\frac{h_{eff}\Delta T_{am}}{3}.$$

## 3. Infrared Temperature Measurement

In this section, after presenting the theoretical foundations of thermography, we describe in detail the different infrared devices used in this work to perform the experimental measurements necessary for the study of the transient regime in the magnetic brake and to verify the validity of the new proposed convective constitutive matrix.

### 3.1. Theoretical Foundations of Thermography

Infrared radiation—IR—is part of the electromagnetic spectrum. It occupies the frequencies between visible light and radio waves and cover wavelengths between 750 nm, the near infrared region, up to 1000 μm, corresponding to the far infrared. This radiation is not visible to the human eye. Like any electromagnetic wave, it travels in a straight line from the source, and it can be reflected, absorbed or it can cross the surfaces of objects that are in its path, depending on the nature of these.

The black body radiation was a challenge for many years. Classical physics did not provide an answer to find a valid formulation in the whole range of wavelengths within the electromagnetic spectrum, corresponding to the infrared. The studies of Kirchhoff, Stefan-Boltzmann, Wien and Planck are of special relevance in this field. It was the latter who solves the problem by introducing a new concept called quantum of energy, giving rise to the birth of Quantum Physics.

Planck’s law of radiation describes the radiation emitted by a black body in thermal equilibrium at a given temperature. It was originally proposed in 1900. This law relates the value of the spectral energy density of the radiation emitted by a black body comprised between the wavelengths *λ* and *λ + dλ*, as a function of the temperature of the body.

Then, Planck establishes for the first time the concept of quantum oscillator. He proposed a natural oscillator of oscillation frequency ν that absorbs or yields a finite amount of energy equal to the product *h·**ν*, where *h* is called Planck constant.

Equation (25) was the first expression proposed by Planck. Subsequently he assigned the values to the constants *C_1_* and *C_2_*, being *C_1_ = 8πhc* and *C_2_ = hc/k_B_*, where *c* the speed of light and *k_B_* the Boltzmann constant:(25)$$I^{b}\left(\lambda ,T\right)d\lambda =\left(\frac{C_{1}}{\lambda^{5}}\right)\frac{1}{\left(e^{\left(\frac{C_{2}}{\lambda T}\right)}-1\right)}d\lambda .$$

When graphically representing the amount of radiant energy emitted per unit of time, per unit of area and per unit of wavelength range as a function of wavelength, a curve is obtained that tends to zero for very short wavelengths and very long. It presents a maximum for a wavelength *λ_max_* that depends on the temperature. If the values of *C_1_* and *C_2_* in (25) are substituted, expression (26) is obtained, see [18]:(26)$$I^{b}\left(\lambda ,T\right)d\lambda =\left(\frac{8\pi hc}{\lambda^{5}}\right)\frac{1}{\left(e^{\left(\frac{hc}{k_{B}\lambda T}\right)}-1\right)}\;d\lambda$$

To find the maximum value of the spectral density of the radiant energy at a given temperature, the Planck equation (26) is derived and is equals to zero. A transcendental equation is obtained in *λ* whose solution is λ_max_ = 0.2014·*hc/kT*. Once the values of the constants *h*, *c* and *k* are substituted, we have the expression (27), which is the well-known Wien displacement law [18]:(27)$$\lambda_{max\;}T=0.002898\;\mathrm{mK}.$$

If the expression (26) is integrated, the Stefan-Boltzmann law [19] is obtained, which relates the total hemispheric emissive power, expressed in W/m^2^, to the absolute temperature of the blackbody by means of the following formula:(28)$$E^{b}=\int_{0}^{\infty }I^{b}\left(\lambda ,T\right)d\lambda =\;\sigma_{SB}T^{4}.$$

In this equation all the wavelengths in which this body radiates are included, and *σ_SB_* has the value 5.67049 × 10^−8^ W⁄m^2^K^4^. Taking into account that there are no ideal black bodies in nature, since these do not exactly comply with the laws described above. However, if it is possible to approximate its behaviour to the black body by making some simplifications.

It can be assumed that a real black body behaves as an ideal one, but that it emits only a fraction of the radiation that an ideal black body could emit under the same temperature conditions and in the same range of wavelengths. In this way, a coefficient called spectral emissivity coefficient *Ɛ* is defined, whose value is between 0 and 1. It indicates how much is the fraction of radiation that can emit a body. It is a property of each material and depends on *λ*:(29)$$I\left(\lambda ,T\right)=\;I^{b}\left(\lambda ,T\right)\varepsilon \left(\lambda \right).$$

Once we have revised this theory, we are able to choose the type of infrared sensor that we need. The temperature to be measured is a function of *λ*. Then we have to search for information about the material on which we want to make the measurements, in order to find the value of its emissivity coefficient.

Applying the Wien displacement law, *λ_max_T* = 2898 μm, we have that for the temperature of interest in our work, whose maximum value is considered 50 °C, equivalent to 323 K, we obtain that the maximum value of the density of energy is produced for a wavelength of 8.97 μm.

As a last step, the suitability of the chosen sensor is checked. In fact, the range of wavelength values in which the Melexis MLX90614 IR sensor works [20], is between 5.5 and 14 μm, which includes the wavelength of 8.97 μm, which produces the maximum intensity of spectral radiation in our experiences, so the choice is correct.

### 3.2. Infrared Sensors

As we have previously said, in this work it is necessary to measure the temperature of a rotating disk without physical contact between thermometer and disk. Also, it is desired that the sensor have a low thermal inertia and that its response is reasonably fast, that it provides a high resolution in its measure and that it is easy to use.

With the above requirements, we have chosen a device from the MLX90614 family of infrared thermometers from the manufacturer Melexis (Ieper ,8900, Belgium), specifically the MLX90614ESF-DCI. This thermometer is of small size, low cost and high precision, which makes it optimal for our measurements.

This infrared detector is a thermopile type and its function is to convert the temperature to the electrical signal that can be processed. A thermopile is a thermal transducer that is formed by several thermocouples connected in series, as shown in Figure 5. The thermocouple is a device that converts radiant energy into an electrical signal. Its principle of operation is based on the Seebeck effect.

The joints between the two metals are called hot junctions and they are the active part of the device. When several thermocouples are joined together to form the thermopile, other connections are formed which are called cold junctions. Hot junctions are those that are exposed to radiation from the body. The cold junctions must be thermally bonded to a surface of the substrate inside the chip and electrically isolated, all maintained at a constant temperature, which is the reference temperature.

This device operates within the temperature range between –40 and 125 °C for the ambient temperature, and between –70 and 382.2 °C for the temperature to be measured. The calibration of this infrared sensor is done in the manufacturing process. It is supplied in a package of standard type TO-39 and its supply voltage is 3 V. This device has a Field of View (FOV) of 5°.

This sensor has an optical band pass filter, which eliminates the effects produced by visible and near infrared radiation, passing only signals whose wavelengths are between 5.5 and 14.4 μm corresponding to the working temperatures of the device. This choice presents an additional advantage because it saves much of the associated electronic circuitry that is needed for the conditioning of the output signal.

The MLX90614 integrates in the same encapsulation two chips, the MLX81101 which is an infrared detector based on the thermopile principle and the signal conditioner ASSP MLX90302 designed to process the output signal of the infrared sensor. Figure 6 shows the block diagram of this device. It illustrates the infrared detector, the signal conditioning circuit and the connection between them. In the block diagram it is observed that the infrared sensor also has an internal thermistor that obtains the temperature of the device and internally compensates the existing temperature gradient.

The operation of the MLX90614 is controlled by a state machine integrated in the chip. This machine synchronizes the data capture of the infrared detector, the low noise signal amplifier, the filters, the analog-to-digital conversion through its 17-bit ADC, the operation of the DSP, and provides data output already processed to other devices using any of the communication protocols.

The DSP is constituted in its hardware part by a microprocessor that has a firmware in EEPROM memory that allows to perform numerical operations at very high speeds with the data obtained in the processes of taking temperatures.

The communication protocols provided in the device are PWM and SMBus. In our case, the communications protocol that has been used is the I2C, since the SMBus of the device is a subset of I2C instructions. The connection of the sensor with the microcontroller— an Arduino Mega 2560—is done through its four terminals in a very simple way.

### 3.3. Thermographic Camera

The same experiments have been carried out using a professional thermographic camera. These experiments have been contrasted with the experimental data obtained with the use of the MLX90614 infrared detectors.

The camera that has been used is a FLIR T-series, model T425 [21] whose characteristics are summarized below. Its resolution is 320 × 240 (134,400 pixels). The spectral range of this camera is between 7.5 and 13 µm. The physical parameters of the material, such as the emissivity coefficient, can be modified easily. We have chosen this model taking into account that it has a measuring range of the target temperature between –20 and 1200 °C, which includes our experimental working range. Its visual field is 25° × 19°/0.4 m.

Sensitivity is a measure of how much the camera can distinguish between small differences in thermal radiation in the thermal image. It is expressed as Noise Equivalent Temperature Difference (NETD) and its unit is K. The value for this camera model is 50 × 10^3^ K which is enough for our measurements. The accuracy of the measurement is ±2 °C or ±2 % of the reading. It has a 3.5" LCD touch screen and incorporates a 3.1 Mpx digital camera for taking photos and videos, as well as software for viewing and processing images using the FLIR Tools.

The images obtained are in JPEG format. They include the data of the measurements made. This camera can be programmed to periodically obtain the temperature of the object and has a removable SD card where the obtained thermographic images can be stored. It has a communication interface through USB 2.0, Bluetooth and Wifi. It also has capacity for video transmission in the infrared.

## 4. Results and Discussion

This section describes the characteristics of the magnetic brake. The numerical validation of the results in the CM is also performed, comparing the results obtained with FEM simulations. In addition, the experimental validation of the CM is carried out using two infrared sensors, the MLX90614 and the FLIR T425infrared camera. The MLX90614 sensor allows point to point readings taken with a higher sampling frequency and the FLIR T425 infrared camera provides a superficial view of the temperatures of the entire disk.

The thermal transient regime of the disk depends on multiple factors. Its origin is in the heat produced by the eddy currents in the disk that are due to its rotation inside a magnetic field generated by a permanent magnet. Therefore, a variable of interest when studying these problems is the speed of rotation of the *w_r_* disk, which influences the effective thermal transmission coefficient *h_eff_*.

The higher the *h_eff_*, the more easily the heat generated in the disk will dissipate and the temperatures in it will be lower. These currents are also influenced by the distance *d* between the magnet and the disk. The greater this distance, the smaller the currents will be and the warmer the disk will be. To give more generality to the work, the experimentation and the simulations have been carried out for different particular cases, with different values of the rotation speed *w_r_* and the distance *d*.

### 4.1. Description and Characteristics of the Magnetic Brake

The magnetic brake consists of a disk of a copper alloy with a volumetric electrical conductivity *σ*_e_ = 4.1 × 10^7^ S / m [13]. The diameter is 315 mm and the thickness is 5 mm. There is a permanent neodymium-iron-boron (NeFeB) magnet located on the disk at a variable distance. The disk is mechanically coupled to the axis of a DC motor by means of six screws, as can be seen in Figure 7. The copper disk has been covered with a very thin film of a black colour plastic material to avoid false readings with the sensors. The surface infrared radiation factor is 0.960. The weight percent composition of the cupper disk is 98 % Cu, 0.82 % Zn, 0.52 % Si, 0.21% Mn, 0.13 % Sn and 0.11 % Al.

The characteristics of the DC motor are the following: AEG, direct current of armature 220 V, 2.2 A, magnetic field powered at 220 V, 0.5 kW, 1400 rpm, IP E22. It can rotate at a variable angular speed *w_r_*, depending on the armature current of the motor, see Figure 7. The distance *d* and the angular velocity are variable depending on the experiment performed. These two magnitudes determine the distribution of heat power in the disk volume, as expressed in the second member of Equation (10).

The permanent magnet, which is used in numerical simulations and laboratory experiments, is modelled in the second quadrant, with magnetic field strength *H* < 0, and remanent magnetism *B_r_* > 0. It is a linear model, with two parameters *B_r_*—permanent magnetism—and μ — magnet magnetic permeability. This model is suitable for permanent magnets of rare-earth elements.

The characteristics of the permanent magnet are the following: the material is NdFeB, brick-shaped, of 50.8 × 50.8 × 25.4 mm, a sense of magnetization according to the axis and on the dimension of 25.4 mm, the coating is Ni-Cu-Ni , manufactured by sintering, the magnetization type is N40, the remaining magnetism *B_r_* is in the interval 1.26-1.29 T, the coercive field strength *H_c_* is in the range 860-955 kA/m, the internal coercive field strength *H_ci_* ≥ 955 kA/m and the maximum energy product is in the range 303-318 kJ/m^3^.

### 4.2. Numerical Validation of the Results in the CM

In Section 4.2.1, we have checked the heat power distribution on the disk using the CM, and we have compared it with FEM. Then, in Section 4.2.2, the influence of the coefficient of effective heat transfer in the transient thermal regime at a point on the disk is analysed.

#### 4.2.1. Heat Power Distribution

The first set of numerical experiments consists of verifying that the second member of the system of equations (10), which correspond to the distribution of heat power due to the currents induced in the disk, are equal applying the CM method and a reference method FEM called GetDP [22] for different values of *w_r_* and *d*.

To do this verification, the system of equations (1) is solved first. Next, the induced currents described in (7) are calculated and, finally, the heat power distribution according to (6) is obtained. The magnet has been placed at a distance *d* = 5 mm above the disk and the disk velocity has varied between 0 and 40 rad/s.

Figure 8 shows the heat power distribution that extends from zero to a maximum value of 6.62 × 10^6^ W/m^3^ for an angular velocity of *w_r_* = 11 rad/s and that occurs at the periphery of the disk. It can be seen that the maximum concentration of heat power is located in the vicinity of the magnet. The mesh consists of 90,593 nodes, 644,741 edges, 1,107,842 faces, and 553,693 tetrahedra.

Figure 9 shows the overall heat power –obtained by adding all the heat power in the disk– as a function of the angular velocity and the increase in heat power with the speed is observed using CM and FEM. Both simulations are practically coincident.

In the same way, experiments are carried out for *d* = 3 mm. In this case, the currents induced in the disk increase due to an increase in magnetic induction. This increases the overall heat power for the same speed, as it is shown in Figure 9. In both cases, *d* = 3 and *d* = 5 mm, the heat powers are coincident, both in the CM and in FEM, which confirms the validity of the new convective matrix formulated for CM.

#### 4.2.2. Analysis of the Thermal Transient Regime

The system of Equation (10) has been solved in the time domain using the Crank-Nicolson scheme:(30)$$\left(\frac{1}{h_{\Delta }}M_{\rho C_{p}}+\theta \left[A\right]\right)\tau^{n+1}=\left(\frac{1}{h_{\Delta }}M_{\rho C_{p}}+\left(1-\theta \right)\left[A\right]\right)\tau^{n}+\left(1-\theta \right){\tilde{W}}^{n}+\theta {\tilde{W}}^{n+1},$$ where:(31)$$A=-DM_{\lambda }G+M_{\rho C_{p}v\;},$$
$h_{\Delta }$ is the increment of time between the steps *n* and *n* + 1, such that θ ∈[0,1] and $\tau^{n}\;$ is the vector of unknowns of the temperatures at time instant *n* associated with the nodes of the domain.

At the initial moment, it starts from a temperature distribution $\tau^{0}$. As a boundary condition, we part from the heat flow associated with the disk surface and the air surrounding the disk, as expressed in (24).

All the equations have been programmed in C ++. The system of Equations (1) and (10) are non-symmetric, as in FEM. We use numerical methods based on the subspaces of Krylov because the matrices are sparse and large. These algorithms are implemented with the numerical package PETSc [23] that uses parallel processing, which reduces the calculation times.

In particular, the linear solver used is the generalized minimal residual algorithm –GMRES. This method is valid for non-symmetric systems, and the absolute and relative tolerances that we have used, in an order of magnitude of 10^−10^, are sufficient to achieve convergence.

We use tetrahedra as elements of the mesh because they are better for complex geometries. The program we use to mesh and visualize the data is the GMSH [24]. As we have said before, the FEM reference program is the GetDP one [22] and we use it to compare and validate our numerical results with the proposed CM method.

For example, for a particular implementation, the execution times in a machine of type Intel core i7-3820, 3.6 GHz and 32 GB of RAM, with four cores and eight threads, is 34 min and 20 s. The transient calculation data are final time *t_f_* = 3000 s, with a time step $h_{\Delta }$ of 0.5 s, a mesh with 83,547 nodes, 589,021 edges, 1,010,764 faces and 505,289 tetrahedra.

Two sets of different simulations have been made: one with a uniform heat power density and the other that parts from the heat power distribution calculated for each CM and FEM obtained from the induced currents.

The first set of numerical simulations is done by setting a uniform heat power density *W^n^* of 2 × 10^6^ W/m^3^ in the area of the disk located under the permanent magnet, in both CM and FEM. That is to say, we do not start from the heat power distribution calculated by each method from the induced currents because this would be slightly different, see Figure 9, and thus no errors are accumulated.

The rotation speed is 15 rad/s, the effective heat transfer coefficient between the disk and the copper is 8.8 W/m^2^K and the total time of the simulation is 2000 s. In Figure 10a the transient regime is represented using the CM and the reference method –FEM–. The control point is the one indicated in Figure 10b with red colour. The temperature distribution shown in the figure corresponds to the instant *t* = 2000 s for the CM. The percentage error obtained from the transient is 0.0029, which turns out to be negligible, as shown in Table 1.

The second set of numerical simulations is represented in Figure 11. In this case, we start from the heat power distribution calculated by each method from the induced currents according to (7) and with the non-uniform heat power distribution corresponding to (6).

In this second case, two types of simulations will be carried out. The transient regime will be calculated in a point as a function of time and, on the other hand, the temperature distribution in a cross section of the disk will be calculated in two determined instants.

Figure 11 shows the variation of temperature, with respect to time, of the point PA of the disk shown in Figure 8. The angular velocity is *w_r_* = 11 rad/s and *d* = 6 mm in all the simulations. The conditions of the thermal transient regime are: time step 0.5 s, total time of the simulation *t_f_* = 800 s. The initial conditions are: τ = 20 °C for the entire disk. And, finally, the ambient temperature *τ_amb_* is equal to 20 °C. The boundary condition consists of varying the effective heat transfer coefficient on all surfaces of the disk *h_eff_*, with the following values: *h_eff_* = [0, 10, 50, 100] W/m^2^K, using equation (23).

Figure 11 shows that as *h_eff_* increases, the final temperature of the permanent regime decreases, as well as the time in which the transient ends. Thus, for example, when *h_eff_* = 50 W/m^2^K the transient ends at 600 s, and for *h_eff_* = 100 W/m^2^K, the transient ends at around 300 s. In the case of perfect insulation, *h_eff_* = 0 W/m^2^K, the temperature rises indefinitely. A permanent regime is not reached. In all the cases studied, the results of the CM analysis coincide with the FEM.

Figure 12a shows the temperature distribution in section AB represented in Figure 12b at time instants *t* = 0, 370 and 790 s, simulated with CM and FEM, with the coefficient of thermal transfer of energy *h_eff_* = 10 W/m^2^K, *w_r_* = 11 rad/s and *d* = 6 mm. The coincidence of the results in both methods is also observed. Based on these simulations, we can conclude that the CM method with the new formulated convective matrix accurately simulates the thermal transient regime in the magnetic brake, as it can be seen in Figure 10a.

### 4.3. Experimental Validation Using Infrared Sensors

Once the validity of the new convective matrix CM for the calculation of the thermal transient regime of the magnetic brake has been demonstrated, we will also carry out an experimental validation using infrared sensors.

We will now compare the experimental data obtained by infrared sensors with CM numerical simulations. In order to apply CM, it is necessary to know exactly the effective value of the thermal transmission coefficient *h_eff_*. Section 4.3.1. shows how an estimation can be obtained analytically and in Section 4.3.2. it is explained how this coefficient can be obtained, in the same way, by means of an adjustment to the experimental data obtained by means of infrared sensors. Then, in Section 4.3.3., it is obtained, by means of two infrared sensors that simultaneously read the temperatures at two characteristic points of the disk, the gradient of the perimeter and radial temperature of the disk. Finally, in Section 4.3.4. the thermograms taken from the disk are compared with CM simulations.

#### 4.3.1. Analytical Calculation of *h_eff_*

The mean convective heat transfer results are nondimensionalised using various dimensionless groups. In this way, they can be applied more universally, for example to machines with different diameters. Two of these groups are the rotational Reynolds number *Re_θ_* and the Nusselt number *N_u_*.

They are defined as follows [25]:(32)$$Re_{\theta }=\frac{w_{r}R^{2}}{\nu },$$
(33)$$N_{u}=\frac{h_{eff}R}{k},$$ where *R* is the rotor radius, *w_r_* is the rotor speed, *ν* is the air kinematic viscosity at ambient temperature, *h_eff_* is the effective convective heat transfer coefficient and *k_a_* is the air conductivity at ambient temperature.

Rotating disks in quiescent air have been studied for many years. Very detailed information can be found in [26]. The mean Nusselt number for laminar flow, is calculated at the external radius as (33):(34)$$N_{u}=a_{1}Re_{\theta }{}^{0.5}.$$

In the case of an isothermal surface, the experimentally/theoretically determined coefficient *a* varies for air depending on the author from *a_1_* = 0.28 [27] to *a_1_* = 0.38 [28]. We have chosen for our calculations a mean value of *a_1_* = 0.33 reported in [29]. In our study the surface of the disk is not isothermal but gradients of temperature are low and we can assume the previous hypothesis. 

Taking into account (32) to (34) we conclude that the mean convective heat transfer coefficient for the disk is:(35)$$h_{eff}=a_{1}k{\left(\frac{w_{r}}{\nu }\right)}^{0.5},$$ with *k* equal to 0.02514 W/mK and *ν* equal to 1.516 × 10^−5^ m/s^2^ at 1 atm and 20 °C [30], and rotor speed *w_r_* of 16.75 rad/s, we obtain a value of mean *h_eff_* of 8.7 W/m^2^K.

#### 4.3.2. Adjusting h_eff_ Using Infrared Technology

In Section 2.4. we explained the boundary condition in the CM and the way to calculate it. In this section, the effective convective heat transfer coefficient between the solid and air *h_eff_* is determined, which better adjusts the simulations to the experimental data and is subsequently validated with new experimental measures.

To estimate the value of *h_eff_*, the MLX90614 infrared temperature sensor is used. The measurement position on the disk is visualized with a laser pointer located at the bottom of the sensor. The axes of the sensor and the laser pointer are vertically separated 11 mm, as illustrated in Figure 13a. This sensor is positioned with a computer numerical control (CNC), in this way the temperature is measured at any given point of the disk, see Figure 13b.

The first set of measurements that we showed has been made at the point called P1 that is on the disk at a distance from the centre x = −140 mm and *y* = 11 mm, as shown in Figure 13b. The conditions of the experiment are: *w_r_* = 16.12 rad/s, *d* = 3 mm and the ambient temperature is 20 °C. The total duration of the experiment is 3000 s –transitory plus permanent regime. The permanent regime is reached at 2000 s.

The ambient temperature in the vicinity of the disk varies due to the energy radiated by the disk, which constantly increases its temperature. The ambient temperature varies very little and has been obtained by means of 16 DS18S20 type thermometers located around the disk. The DS18S20 [31] is a programmable digital thermometer with resolution from 9 bits to 12 bits. It communicates over to One-Wire bus with a central microprocessor. Measures temperatures are from -55 °C to +125 °C ± 0.5 °C, accuracy goes from -10 °C to +85 °C, with 9-bit resolution.

The experimental data of the surface temperature and the ambient temperature are represented by triangles and points, respectively, in Figure 14. The results of the simulations with the cell method, –CM1, CM2 and CM3– are represented by the continuous lines of different thicknesses.

On the disk there are two clearly differentiated zones, a solidary zone with the axis of the DC machine, whose material is iron and has a radius of 43 mm, and the other of copper that is the rest of the disk. We have considered, therefore, two effective coefficients of heat transfer denominated *h_eff_*
_(Cu)_ and *h_eff_*
_(Fe)_ for the area of ​​copper and iron, respectively.

Different combinations have been tested for the values of these coefficients, obtaining the simulations CM1 [*h_eff_*
_(Cu)_ = 25, *h_eff_*
_(Fe)_ = 32], CM2 [*h_eff_*
_(Cu)_ = 50; *h_eff_*
_(Fe)_= 32], CM3 [*h_eff_*
_(Cu)_ = 8.8; *h_eff_*
_(Fe)_= 32] W/m^2^K that are represented in Figure 14. It is observed that as the result that best fits the simulation and the experimental measure corresponds to the simulation CM3.

As it can be observed, the coefficient *h_eff_*
_(Cu)_ = 8.8 W/m^2^K in the copper zone is precisely adjusted to the value of 8.7 W/m^2^K obtained in Section 4.3.1. The coefficient that best fits the data of the simulations with the experimental data in the iron zone is *h_eff_*
_(Fe)_ is 32, which is higher than that of copper, because it takes into account the heat losses that are produced by conduction in the DC machine, and that are included in this coefficient.

In order to do an additional verification, new experiments have been carried out at another point on the disk, called P2. This is placed on the disk at a distance from the centre *x* = −110 mm and *y* = 11 mm. The separation of the disk to the magnet is *d* = 3 mm and *w_r_* = 15.7 rad/s. The ambient temperature and that of this point are measured with the Pt100 temperature sensor included in the infrared sensor MLX90614.

The results obtained are summarized in Figure 15. The parameters that best fit the simulation data with the experimental data correspond to *h_eff_*
_(Cu)_ = 8.4, *h_eff_*
_(Fe)_ = 30 W/ m^2^K. They are slightly lower than those obtained in the previous section because the rotation speed is slightly lower.

In order to have a global view of the distribution of the temperature on the disk surface, the results are shown in Figure 16 also in a 3D representation.

#### 4.3.3. Gradient of the Perimeter and Radial Temperature Measured with Two Infrared Sensors

In all the simulations carried out using the CM, a gradient of the perimeter temperature in the direction of rotation is observed, as shown in Figure 10 and Figure 12. To verify these results, two new experiments were carried out with two IR sensors of the MLX90614 type. In this way it is possible to simultaneously measure the temperature in those two points.

These sensors are oriented towards two points, one located right next to the magnet called Pu and another located in a diametrically opposite position called Pd, as shown in Figure 17. The sensors are at a distance of 10 cm from the surface of the disk. The coordinates of the point Pu on the surface of the disk are *x* = 111 mm and *y* = 80 mm and those of the point Pd are *x* = −105 mm and *y* = −78 mm.

A first experiment was performed at a disk rotation speed *w_r_* = 15.28 rad/s, in the counter clockwise direction, the electrical power consumed by the DC motor is 67 V × 2.1 A = 140 W. The initial temperature of the entire disk is 17.35 °C and the ambient temperature is 17.3 °C. The total duration of the experiment is 5400 s and the permanent regime is reached in 2700 s, with a temperature of 41 °C. The sampling of data in each sensor is done every second and the results are shown in Figure 18b. We observe that the average temperature difference between these two points, located near the perimeter of the disk, is 0.39 °C.

A second experiment was performed at a rotation speed *w_r_* = 10.89 rad/s, in the counter clockwise direction. The electrical power consumed by the direct current motor is 49.6 V × 1.52 A = 75.39 W. The initial temperature of the entire disk is 17.31 °C, the ambient temperature is 17.8 °C. The total duration of the experiment is 5400 s. The permanent regime is reached in 2700 s, see Figure 18a, with a temperature of 32 °C that is lower than that of the first experiment, see Figure 18a. An average temperature difference between these two points of 0.24 °C is observed, which is 0.15 °C lower than the first experiment due to its lower rotation speed, see Figure 18b.

Figure 19 shows the results corresponding to the temperatures at points Pu and Pd, using the CM, and those corresponding to the experimental data of Figure 18b. The results of the simulation are better adjusted in the permanent regime and the initial part of the transient. The parameters used for the adjustment in the simulation are [*h_eff_*
_(Cu)_ = 8.2, *h_eff_*
_(Fe)_ = 27.4] W/m^2^K.

Secondly, to finish, the radial gradient of temperatures on the disk is studied. The temperatures are compared in two superficial points, Pc and Pu, located in the centre and in the periphery, respectively, as can be seen in Figure 20. The experiment was carried out at a disk rotation speed *w_r_* = 16.33 rad/s, in the counter clockwise direction.

The initial temperature of the entire disk is 17.26 °C and the ambient temperature is 19.0 °C. The total duration of the experiment is 3500 s. The permanent regime is reached, in 2500 s with a temperature in the periphery of 43 °C and in the centre of 36 °C, as it is observed in the Figure 21. In the same way we have represented the CM simulations with fitting parameters *h_eff_*
_(cu)_ = 8.2, *h_eff_*
_(fe)_ = 32 W/m^2^K.

Taking into account the comparison of the experimental data obtained by infrared sensors and the results of the numerical simulations we can conclude that CM adequately represents the thermal transient regime of the brakes through the new convective constitutive matrix proposed in this article and can be used in the design of these brakes.

#### 4.3.4. Temperature Distribution on the Surface of the Disk, Obtained by the Infrared Camera

The measurements made with the MLX90614 infrared thermometers are suitable for measuring the temperature at one point. To simultaneously analyse the transient temperature regime over the entire surface of a disk, a temperature measurement in a two-dimensional point array is needed.

The measurement of the two-dimensional distribution of the temperatures on the surface of the rotating disk is made with the FLIR camera model T425. The thermograms have been taken with a programmed emissivity ε = 0.96, relative humidity 82% and the distance from the camera to the disk is 1 m. The arrangement of the camera and a thermogram taken with it can be seen in Figure 22 and Figure 23. An experiment was carried out with the following conditions: *d* = 3 mm, *w_r_* = 16.54 rad/s and ambient temperature 18 °C. It has been proven that in 2000 s the permanent thermal regime is reached. The disk is attached to the shaft of the electric machine with six screws. Their heads protrude from the disk and therefore they are cooled more easily, being their temperature lower as shown in all the thermograms, see Figure 24.

Figure 24a shows the thermogram measured with the FLIR T425 camera at the instant 245 s. Two cuts Cut1 and Cut2 are represented, which extend from the centre to the perimeter of the disk. The purpose of these cuts is to observe the variation of the temperature from the centre to a hotter zone, Cut1, and from the centre to a colder zone, Cut2. In Figure 24b this thermogram is compared with those obtained by CM.

The temperature distribution is represented in 3D, to highlight the hottest area near the permanent magnet, on the perimeter of the disk, as shown in Figure 25. By the decrease in temperature that is seen in this figure, we can deduce the direction of rotation. The disk rotates counter clockwise.

In all the thermograms appear two black rectangles that correspond to the connectors of a contactor of the electric machine and the support of the permanent magnet, which are at ambient temperature.

Figure 26 shows the temperatures along the parametric cuts Cut1 and Cut2, as a function of the distance to the centre. It is observed that throughout the cut, the temperature in Cut1 is higher than that obtained in Cut2, as expected. It is also observed, in the graph Cut2, a relative minimum in the temperature that corresponds to the temperature of the screw clamping the disk.

In Figure 27a a new distribution of the temperature in the whole disk is represented for an instant of time of t = 1565 s. A diametric cut is made, according to the direction indicated in this figure. In Figure 27b this thermogram is compared with those obtained by CM. The results of this cut are shown in Figure 28. Three relative minima are observed corresponding to two screws clamping the disk and the centre of the disk attached to the motor shaft. Again identical conclusions are observed as in the figure corresponding to the time instant t = 245 s.

To estimate the discrepancy between FEM and CM simulations and the experimental results obtained using IR techniques, the metrics reflected in Table 1 and Table 2 have been studied.

The determination coefficient (R^2^) values indicate a good adjustment of the data in all the comparatives. The values of the root mean square perceptual error (RMSPE), mean absolute percentage error (MAEP) and percentage bias (PBIAS) are relatively high in the comparative C1. It must be taken into account that they are global heat power values, in watts. The explanation may be, probably, in the small number of contrasted cases. This is due to the difficulty of performing multiple simulations due to the amount of time required. Even so, all the indicators are in the optimal range. Looking at Table 1, we can assure that the biggest error committed is the average percent squared error, RMPSE, whose value is 0.0459%. It is a more than acceptable value.

## 5. Conclusions

In this work we have analysed the transient regime of the temperature distribution of a rotating conductor disk within a stationary magnetic field generated by a permanent magnet that induces electrical currents. The system acts as a magnetic brake. For the simulation and analysis of this type of problems, a new thermal convective matrix has been formulated in the 3D Cell Method. Thanks to the experimentation carried out by infrared sensor instrumentation, it is concluded that the Cell Method with the formulation of a new convective matrix is suitable to simulate thermal problems such as the design of magnetic brakes. Simulations using FEM also confirm the validity of this new method. The difference between the experimental results obtained by infrared sensors and those obtained in the Cell Method and FEM numerical simulations are lower than 0.0459%.

## Figures and Tables

**Figure 1 sensors-19-02028-f001:**
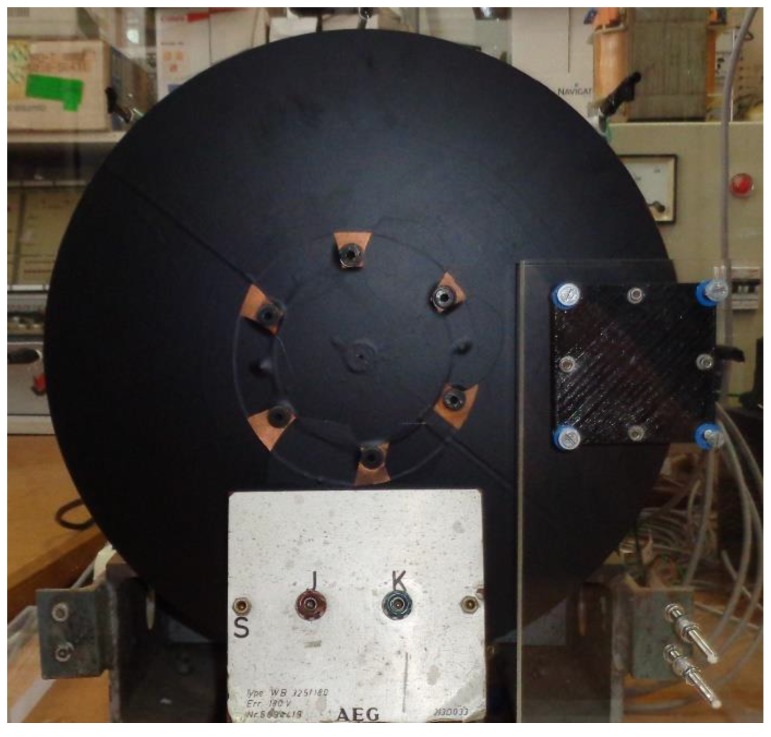
Front view of the disk and location of the thermometers installed to measure the ambient temperature around the disk.

**Figure 2 sensors-19-02028-f002:**
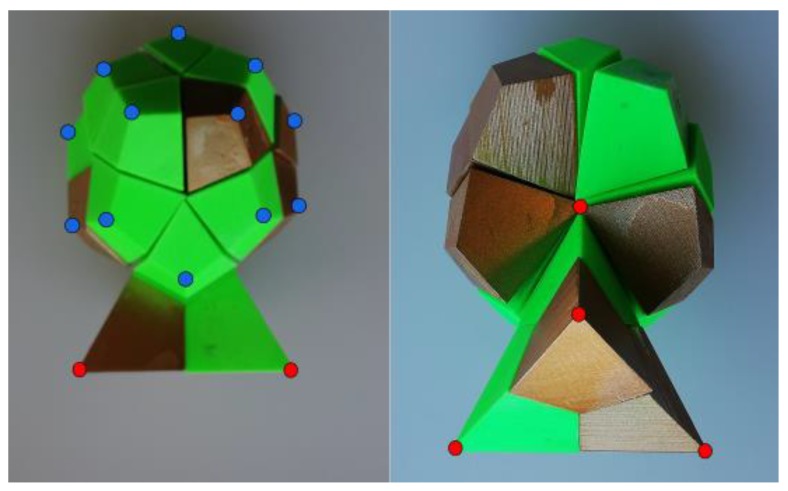
Volume of the dual space obtained as a union of portions of the tetrahedron of the primal mesh by barycentric division. Nodes of the dual mesh–blue colour– and nodes of the primal mesh–red colour.

**Figure 3 sensors-19-02028-f003:**
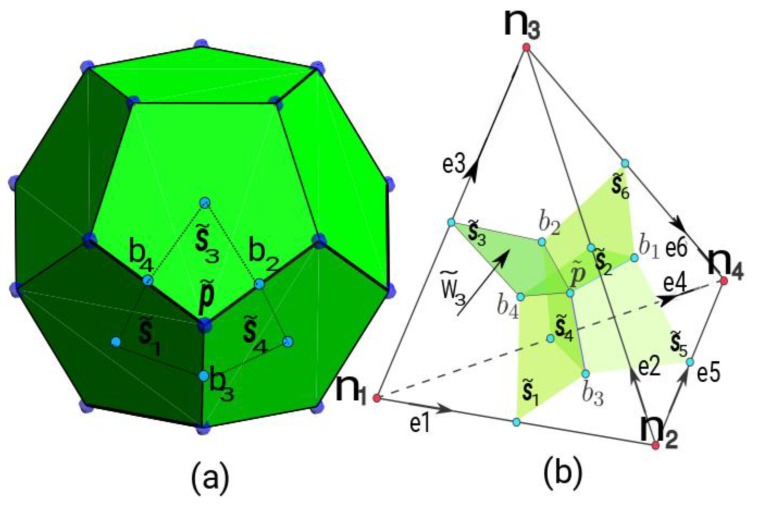
(**a**) Dual volume $\tilde{V}$ and nodes of the dual volume in blue; (**b**) Primal volume *V*, primal nodes *n* in red with oriented primal edges *e*, and dual planes $\tilde{S}$ in green.

**Figure 4 sensors-19-02028-f004:**
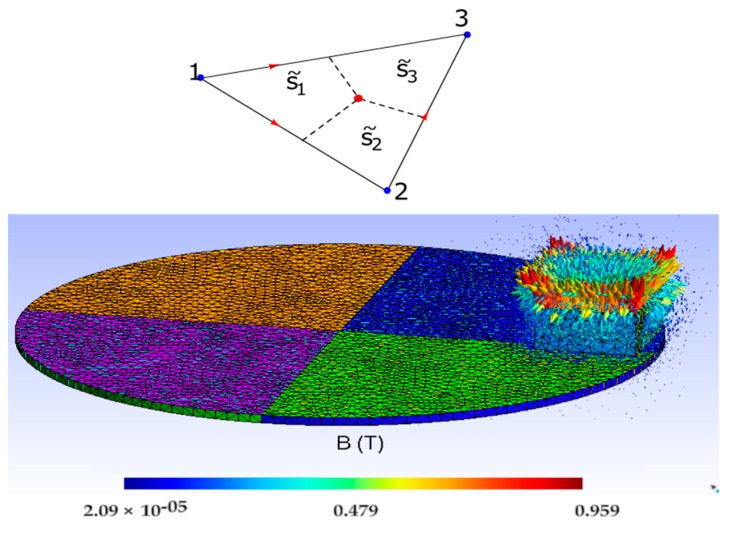
Triangle of reference 1, 2, 3 and dual surfaces ${\tilde{S}}_{1},\;{\tilde{S}}_{2},\;{\tilde{S}}_{3}$. Contour triangles on the disk.

**Figure 5 sensors-19-02028-f005:**
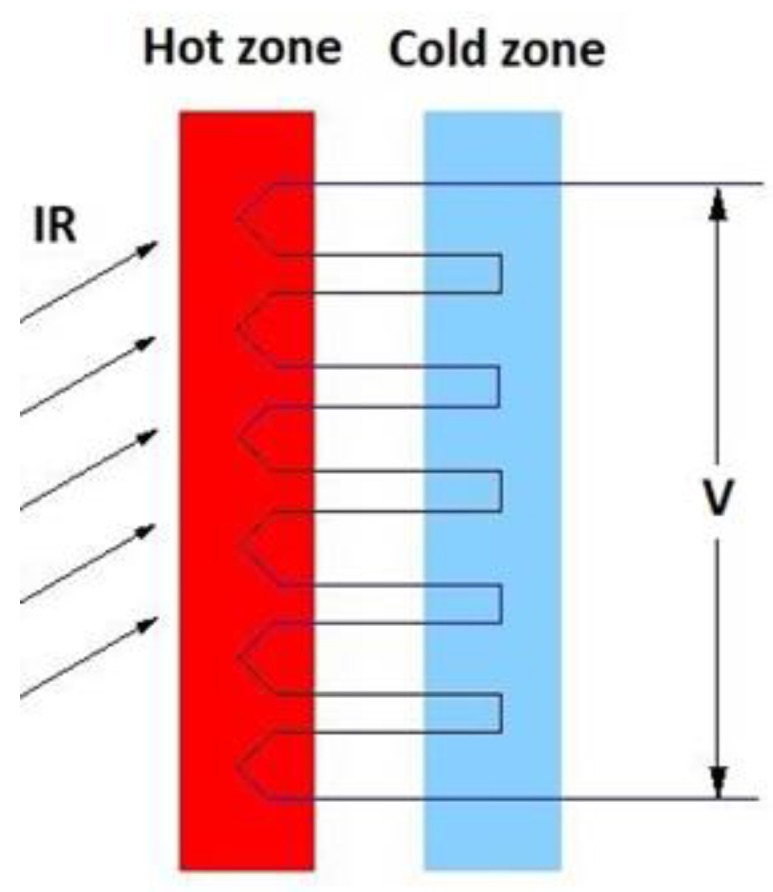
Scheme of a thermopile.

**Figure 6 sensors-19-02028-f006:**
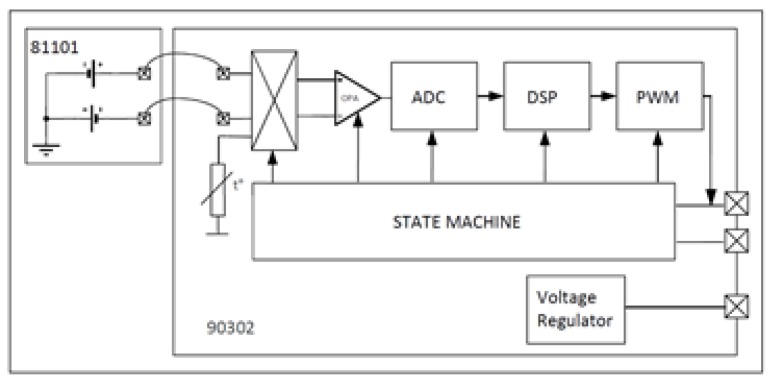
Block diagram of MLX90614.

**Figure 7 sensors-19-02028-f007:**
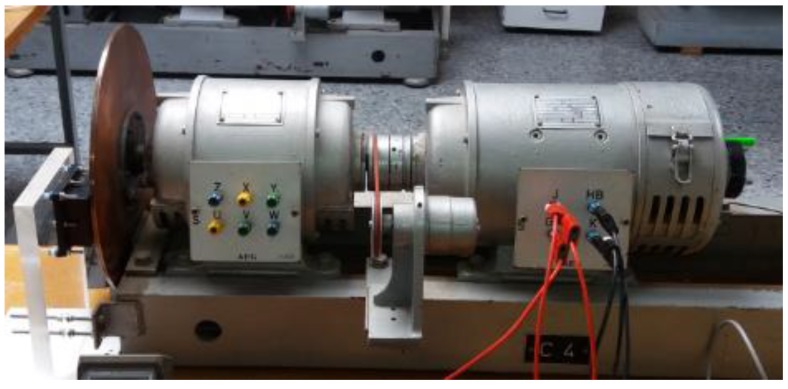
On the left the permanent magnet and copper disk, on the right the DC motor to modify the angular velocity *w_r_*.

**Figure 8 sensors-19-02028-f008:**
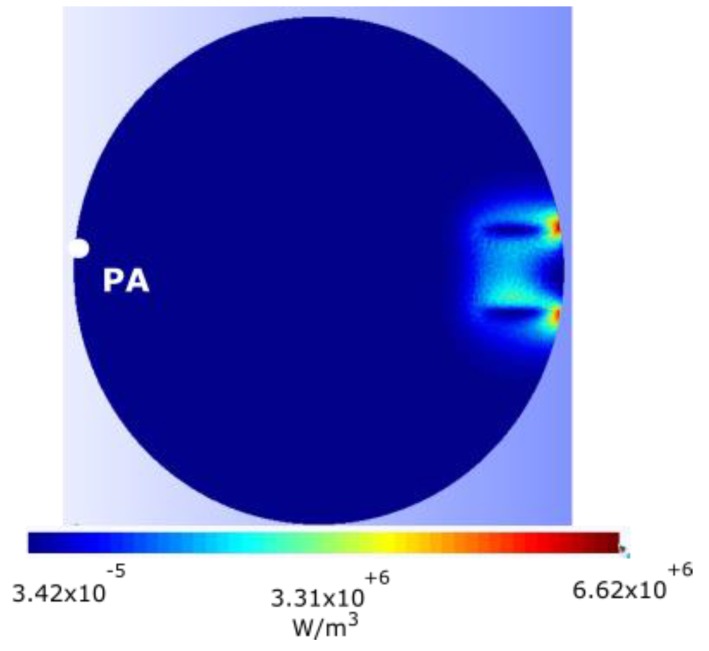
Distribution of volumetric density of the heat power in the disk, due to induced currents with *w_r_* = 11 rad/s and *d* = 5 mm, with a total heat power of 32.26 W.

**Figure 9 sensors-19-02028-f009:**
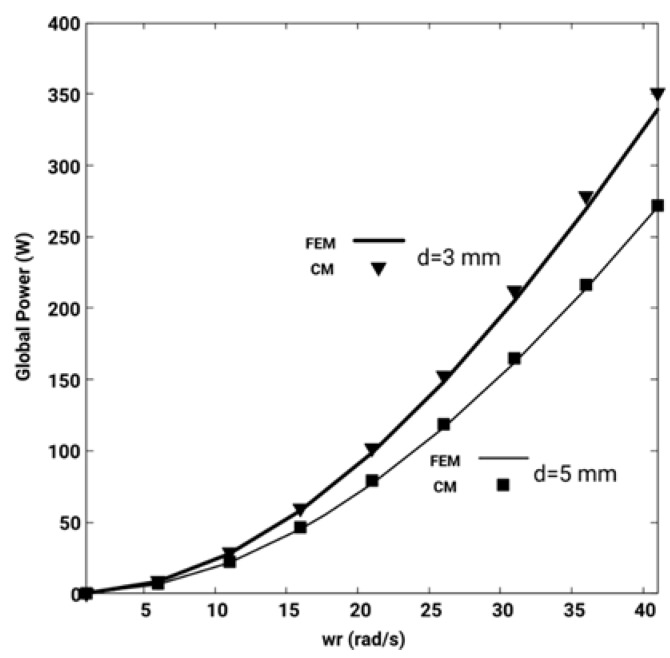
Comparison of the global heat power in the disk as a function of *w_r_*, for *d* = 3 and 5 mm, using the CM and the FEM. Both simulations are practically coincident.

**Figure 10 sensors-19-02028-f010:**
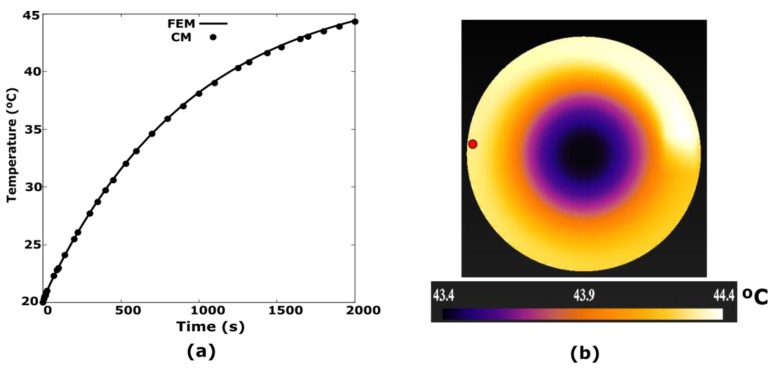
(**a**) Transient temperature regime at the red point, using CM and FEM with a uniform heat power density of 2 × 10^6^ W/m^3^ and *w_r_* = 15 rad / s; (**b**) The surface temperature distribution shown corresponds to *t* = 2000 s for the CM.

**Figure 11 sensors-19-02028-f011:**
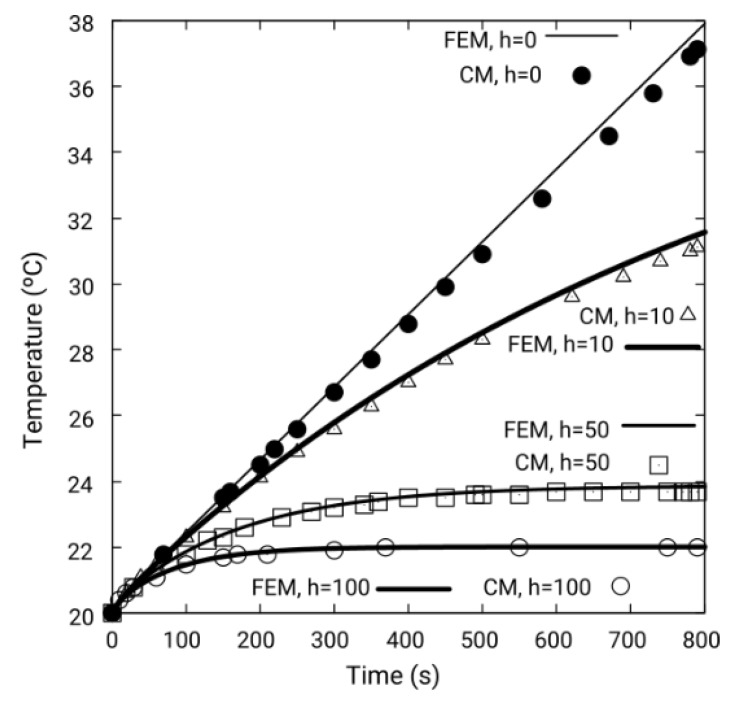
Comparison of the transient temperature regime in point PA of Figure 8 for *h_eff_* = 0, 10, 50 and 100 W/m^2^K, using CM and FEM, for *w_r_* = 11 rad/s and *d* = 6 mm, for a non-uniform heat power distribution. The results by both methods are practically coincident.

**Figure 12 sensors-19-02028-f012:**
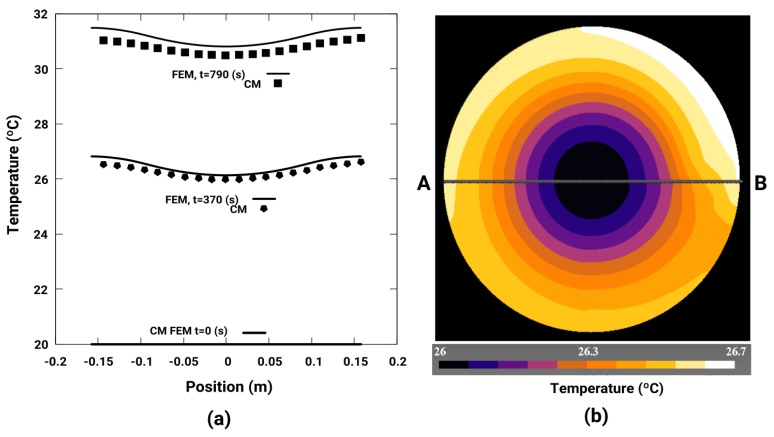
(**a**) Temperature distribution in section A-B at instants *t* = 0, 370 and 740 s for CM and FEM; (**b**) Surface distribution of the temperature at time *t* = 370 s for *h_eff_* = 10 W/m^2^K, *w_r_* = 11 rad / s and *d* = 6 mm, calculated with CM.

**Figure 13 sensors-19-02028-f013:**
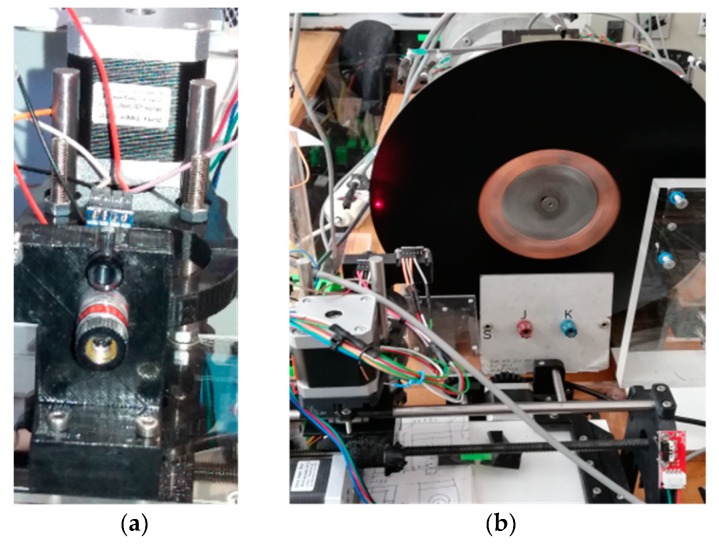
(**a**) Detail of the infrared temperature sensor MLX90614 and laser pointer; (**b**) CNC to position the infrared sensor and measure the temperature at different points on the disk.

**Figure 14 sensors-19-02028-f014:**
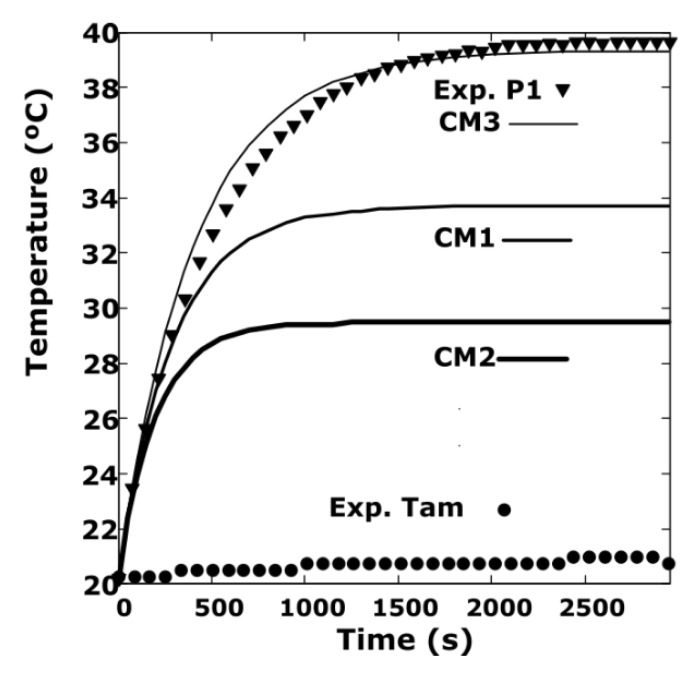
Adjustment of the simulation of the transient regime at point P1, compared to the experimental data, CM1, CM and CM3 with *d* = 3 mm and *w_r_* = 16.12 rad/s. The ambient temperature –Exp. Tam– and the temperature of point P1 –Exp. P1– are measured with the MLX90614 temperature sensor.

**Figure 15 sensors-19-02028-f015:**
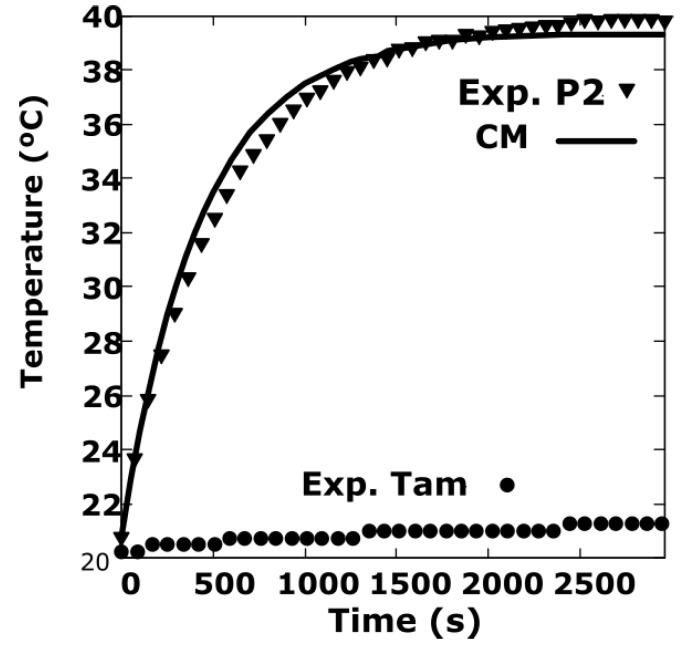
Verification of the simulation of the transient regime at point P2 versus the experimental data with the following values: CM [*h_eff_*
_(Cu)_ = 8.4, *h_eff_*
_(Fe)_ = 30] W/m^2^K, *d* = 3 mm y *w_r_* = 15.7 rad/s. The ambient temperature and that of point P2 are measured with the MLX90614 temperature sensor.

**Figure 16 sensors-19-02028-f016:**
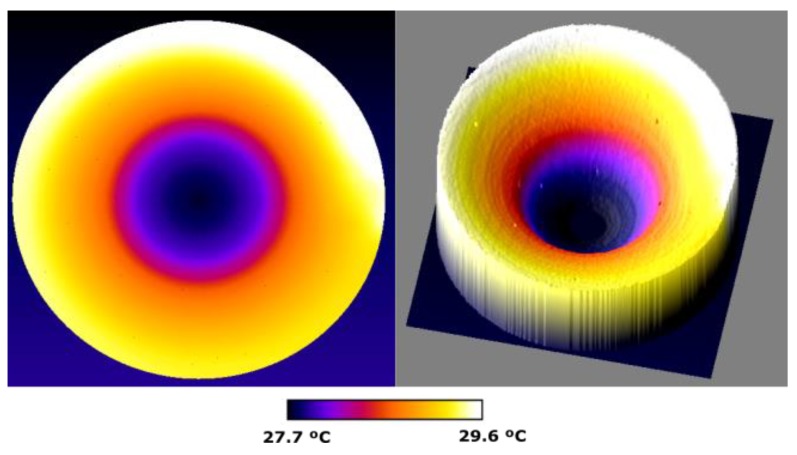
3D representation of the temperature corresponding to the simulation with the CM at the instant of 250 s, *h_eff_*
_(Cu)_ = 8.4, *h_eff_*
_(Fe)_ = 30 W/m^2^K, *d* = 3 mm and *w_r_* = 15.7 rad/s.

**Figure 17 sensors-19-02028-f017:**
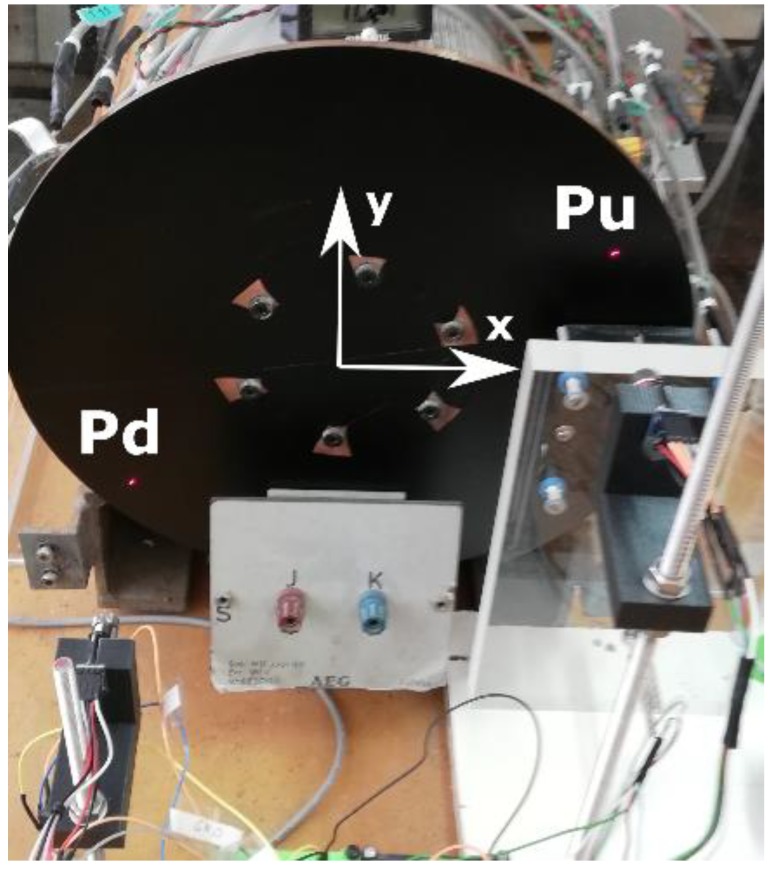
Perimeter measurement of the temperature simultaneously at two points, Pu and Pd, with two infrared thermometers MLX90614.

**Figure 18 sensors-19-02028-f018:**
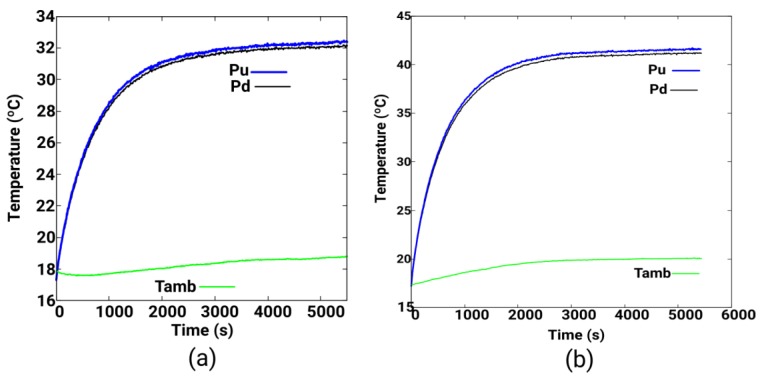
(**a**) Perimeter temperature measurement with two infrared thermometers MLX90614 at Pu and Pd, and ambient temperature, with *w_r_* = 10.89 rad/s; **(b**) And with *w_r_* = 15.28 rad/s.

**Figure 19 sensors-19-02028-f019:**
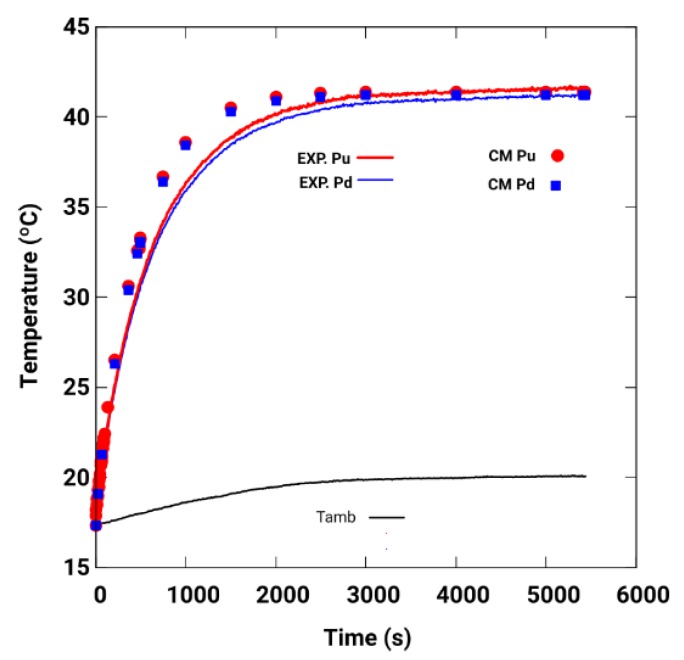
CM perimeter simulations with adjustment parameters *h_eff_*
_(Cu)_ = 8.2, *h_eff_*
_(Fe)_ = 27.4 W/m^2^K in Pu and Pd, and experimental data Exp. Pu and Exp. Pc, for *w_r_* = 15.28 rad/s.

**Figure 20 sensors-19-02028-f020:**
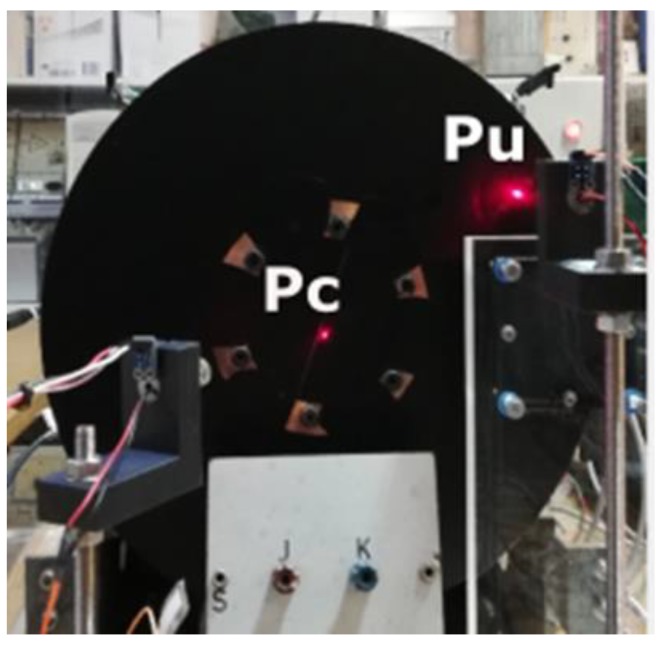
Situation of the points Pc and Pu, located in the centre and in the periphery, respectively, where the radial gradient of temperatures in the disk is measured.

**Figure 21 sensors-19-02028-f021:**
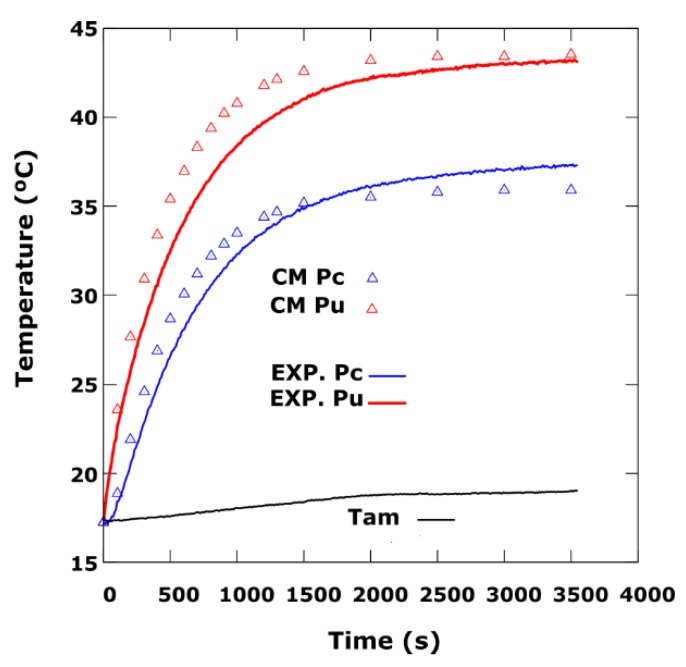
Radial simulations of the CM with adjustment parameters *h_eff_*
_(Cu)_ = 8.2, *h_eff_*
_(Fe)_ = 32 W/m^2^K, and experimental curves obtained in Pu and Pc for *w_r_* = 16.33 rad/s.

**Figure 22 sensors-19-02028-f022:**
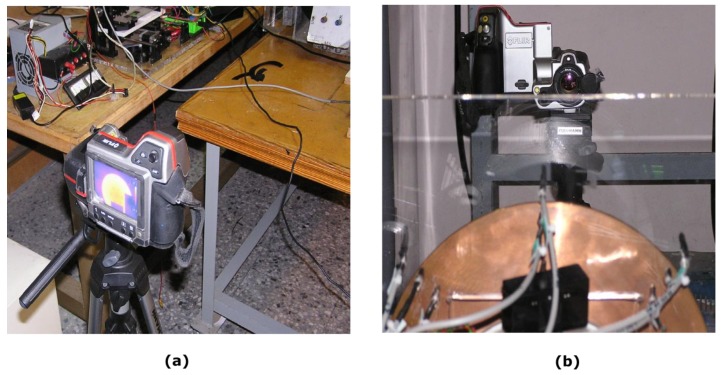
(**a**) Rear view of the FLIR T425 camera; (**b**) Front view of the FLIR T425 camera.

**Figure 23 sensors-19-02028-f023:**
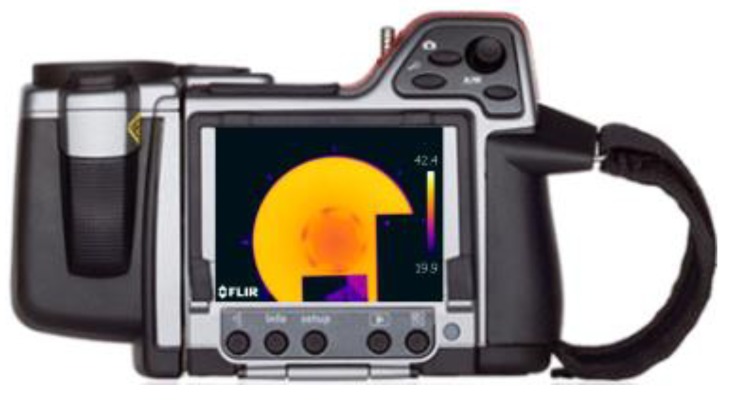
Thermogram taken with the FLIR T425 camera at instant 1565 s.

**Figure 24 sensors-19-02028-f024:**
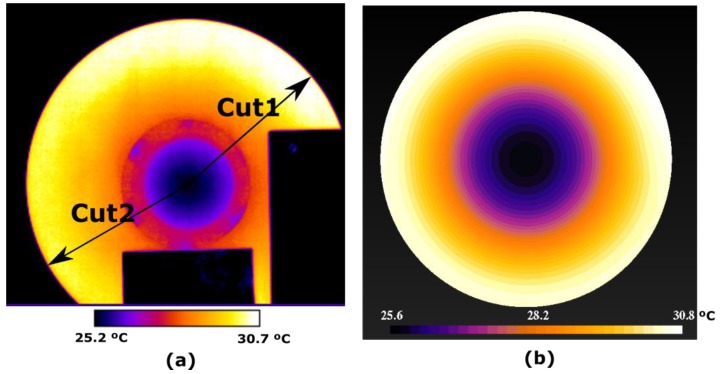
(**a**) Thermogram taken with the FLIR T425 camera at the instant 245 s compared with the simulation obtained under the same conditions by CM in (**b**), with *d* = 3 mm. The parametric cuts Cut1 and Cut2 are shown.

**Figure 25 sensors-19-02028-f025:**
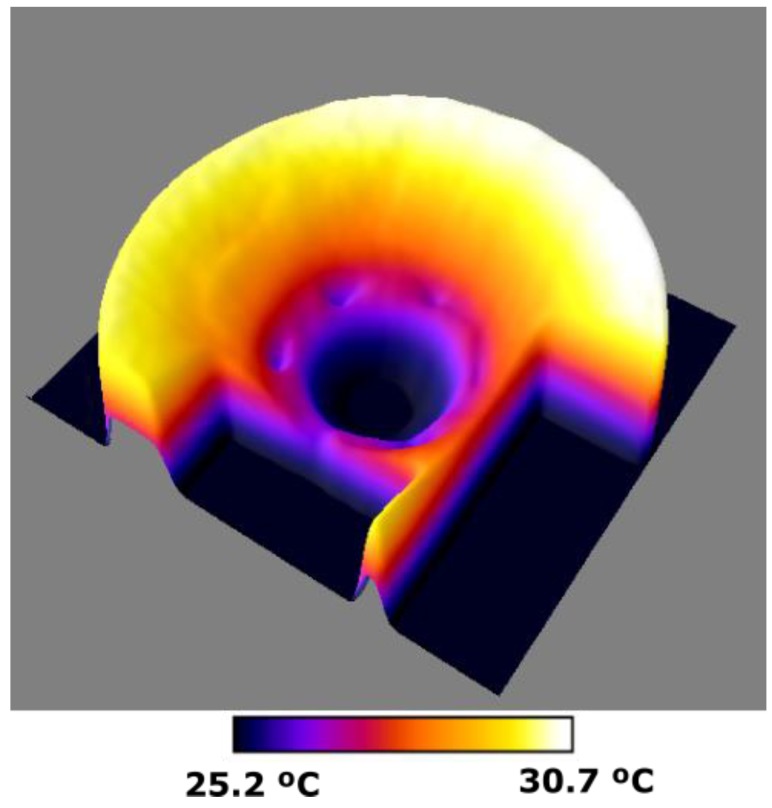
3D representation of the thermogram taken with the FLIR T425 camera of Figure 25. The coldest points of the screws that join the disk to the axis of the machine are observed.

**Figure 26 sensors-19-02028-f026:**
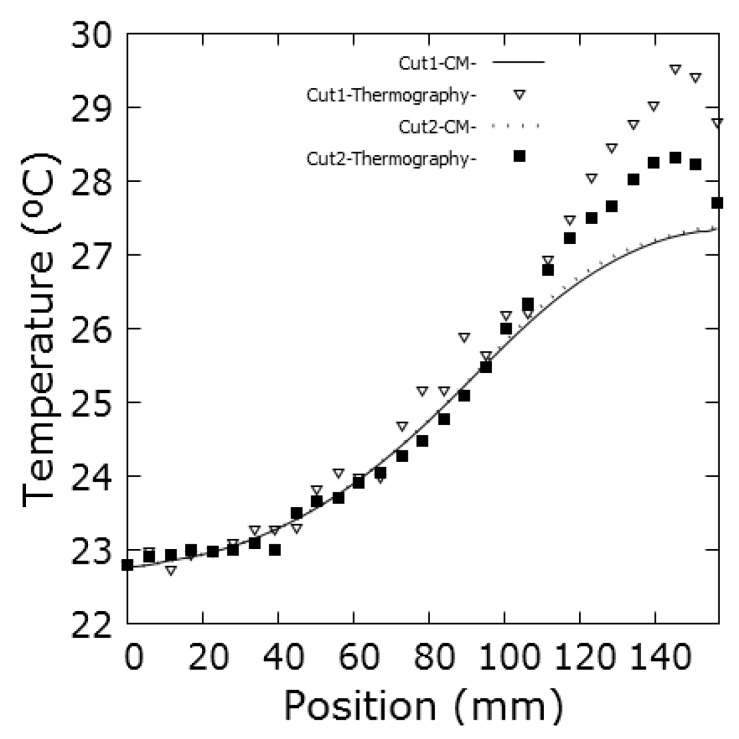
Temperatures obtained by CM in the parametric sections –Cut1 and Cut2– from the centre of the disk to the periphery, at the instant of 245 s, see Figure 24b. The thermogram has been taken with the FLIR T425 thermal camera and the thermographic data has been obtained with the Fiji image analyser [32], see Figure 24a.

**Figure 27 sensors-19-02028-f027:**
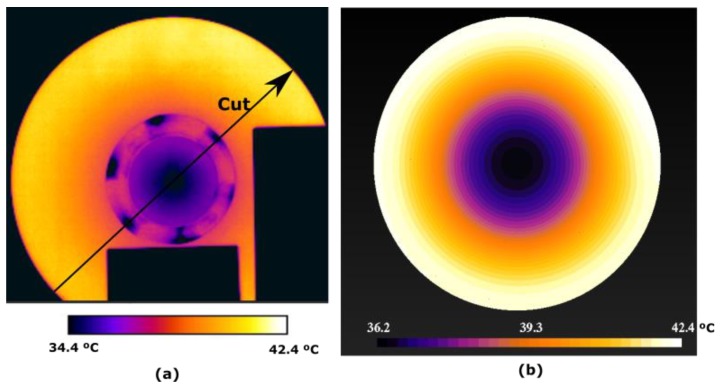
(**a**) Parametric diametric cut –Cut– of the thermogram at instant 1565 s, compared to a CM simulation (**b**) made under the same conditions. Measurement made with the FLIR T425 camera.

**Figure 28 sensors-19-02028-f028:**
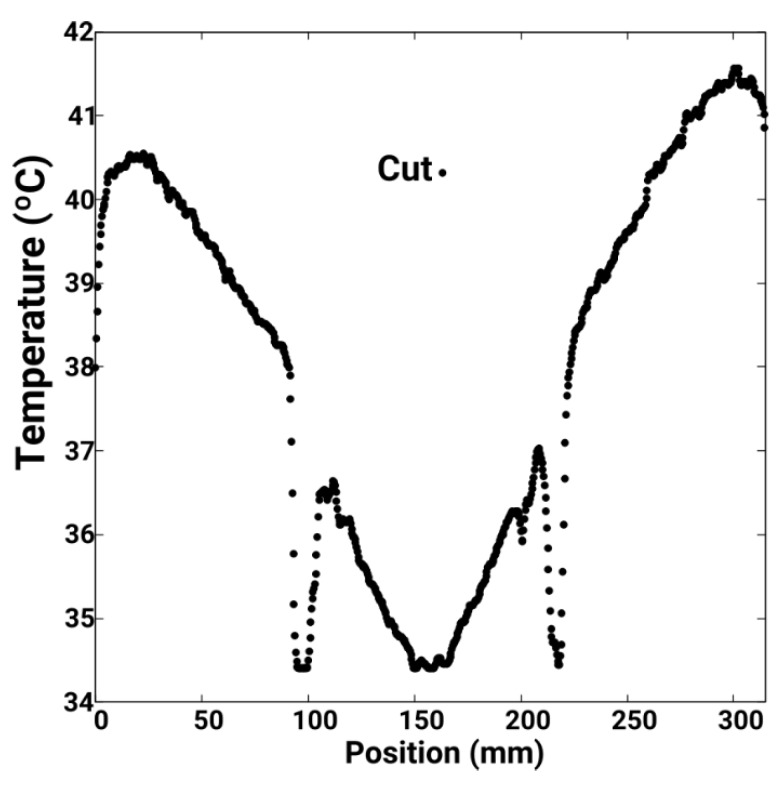
Temperature distribution of the parametric cut of Figure 27a, at instant 1565 s. The thermographic data has been obtained with the Fiji image analyser.

**Table 1 sensors-19-02028-t001:** Metrics of the comparisons proposed.

Comparison	C1	C2	C3	C4	C5	C6	References
R^2^ [0, +1] Optimum: +1	1.0000	0.9999	1.0000	0.9999	0.9848	0.9733	[33]
RMSPE [−1, +1] Optimum: 0	0.0459	0.0146	0.0029	0.0024	0.0183	0.0365	[34]
MAEP [−1, +1] Optimum: 0	0.0342	0.0119	0.0024	0.0020	0.0125	0.0238	[34]
PBIAS [−1, +1] Optimum: 0	0.0330	0.0111	0.0022	0.0020	0.0085	−0.0234	[35]

**Table 2 sensors-19-02028-t002:** Some numerical experiments developed.

**C1:**	Comparison of the global heat power in the disk as a function of *w_r_*, for *d* = 3, using the CM and the FEM. See Figure 9.
**C2:**	Comparison of the global heat power in the disk as a function of *w_r_*, for *d* = 5 mm, using the CM and the FEM. See Figure 9.
**C3:**	Transient temperature regime at the red point, using CM and FEM with a uniform heat power density. See Figure 10a.
**C4:**	Verification of the simulation of the transient regime at point P2 versus the experimental data. See Figure 15.
**C5:**	Temperatures obtained by CM in the parametric section Cut1. See Figure 26.
**C6:**	Temperatures obtained by CM in the parametric sections Cut2. See Figure 26.

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
