# Peer review of "Thermal Analysis of a Magnetic Brake Using Infrared Techniques and 3D Cell Method with a New Convective Constitutive Matrix"

_sensors, 2019, doi:10.3390/s19092028_

Reviewer 1 Report

The paper is a continuation of the research line of [13].

Concerning the constitutive matrices for the Cell Method, which is the main novelty of the paper, there has been a new technique that you may consider in the future, see https://ieeexplore.ieee.org/stamp/stamp.jsp?arnumber=8636551.

The eddy current problem and the thermal problem are solved in a decoupled way. Is this because the time constants are different? Would it make sense to solve the two problems tightly coupled?

There is no mention about the solver used to solve the ODE system (5). Please indicate the library or at least the technique used (Euler explicit or implicit, Runge-Kutta, Crank-Nicolson, etc...).

From (17) it appears that the new constitutive matrix is not symmetric. As far as I know it is the same in other numerical methods like Finite Elements. Does the fact that the matrix is not symmetric causes troubles in the numerical solution of the problem? Again, what solver has been used for solving the transient thermal problem?

What is completely missing is a discussion on what is the advantage of using the cell method with respect to a consolidated commercial software like COMSOL or research software like GetDP? Flexibility? Speed? Confidence in the results (because no "tricks" have been used inside the software)?

Please comment why the MLX90614ESF-DCI sensor has been used in addition to a standard thermal camera. Is it because of cost? Or other things like required speed or other variables. If using a commercial thermal camera *only*, probably the whole section III can be shorted a lot. If there are no other reasons, it seems to me that get the temperature through a termal camera is standard and does not require theory behind it.

After going further with the paper I did not found a comparison of the two temperature sensors, I wander why. Maybe it would be better to say from the beginning (i.e. in the introduction) that two different methods will be used and carefully explain why two methods to measure the temperature are necessary.

Kryslov should be Krylov.

I found many information I asked before in the review in the result section. Perhaps it would be desirable to advise the readers in Section 2 that these details will be presented later in Section 4.

Author Response

REVIEWER 1

Open Review

English language and style

( ) Extensive editing of English language and style required
(x) Moderate English changes required
( ) English language and style are fine/minor spell check required
( ) I don't feel qualified to judge about the English language and style

As we carefully checked the spelling, we have noted that some paragraphs had the Word corrector in UK English and other in USA English. We have selected the whole document and selected UK English. As a consequence, we have changed a few words in the document as analyse, centre and colour that are highlighted in yellow in the paper on lines 13, 16, 38, 128, 132, 133.

On line 428 we had used a different font for “neodymium” and we have corrected it.

Finally, on line 493 were we said “Kryslov” it should be “Krylov”.

Once we have read in detail the paper we have found two grammatical mistakes. The grammatical construction is the same and follows a subjunctive English structure. So we have changed “has” for “have”, both on lines 63 and 338.

We have also changed Kelvin grades, “°K”, by Kelvin “K” in all the document because Kelvin is not a graduated scale as it is Celsius scale, “°C”.

Yes

Can be   improved

Must be   improved

Not   applicable

Does the   introduction provide sufficient background and include all relevant references?

( )

(x)

( )

( )

Is the   research design appropriate?

(x)

( )

( )

( )

Are the   methods adequately described?

(x)

( )

( )

( )

Are the   results clearly presented?

(x)

( )

( )

( )

Are the   conclusions supported by the results?

(x)

( )

( )

( )

Comments and Suggestions for Authors

The paper is a continuation of the research line of [13].

Yes. In this paper we study the thermal problem, which was not studied in [13].

Concerning the constitutive matrices for the Cell Method, which is the main novelty of the paper, there has been a new technique that you may consider in the future, see https://ieeexplore.ieee.org/stamp/stamp.jsp?arnumber=8636551. 

Thank you. It is a very interesting paper. We will keep it in mind in our future work.

The eddy current problem and the thermal problem are solved in a decoupled way. Is this because the time constants are different? Would it make sense to solve the two problems tightly coupled?

The eddy current problem and the thermal problem are solved in a decoupled way because the physical material properties in eddy current problem are constant in the range of temperature in which we are working. But, the reason is not that the time constants are different, because the magnetic field is stationary and the angular velocity is constant, and, hence, eddy current are constant along the time. So, it does not make sense to solve the problems tightly coupled.

There is no mention about the solver used to solve the ODE system (5). Please indicate the library or at least the technique used (Euler explicit or implicit, Runge-Kutta, Crank-Nicolson, etc...). 

In Section 4.2.2. we cite in an explicit way that we have programed in C++ the method Crank-Nicolson to solve the ODE system (10) that includes the convective term, as it is shown in equation (30).

From (17) it appears that the new constitutive matrix is not symmetric. As far as I know it is the same in other numerical methods like Finite Elements. Does the fact that the matrix is not symmetric causes troubles in the numerical solution of the problem? Again, what solver has been used for solving the transient thermal problem?

You are right. The new constitutive matrix (17) is not symmetric as in other numerical methods like Finite Elements. The fact that the matrix is not symmetric does not cause troubles in the numerical solution of the problem because we have used an adequate solver. As we have explained in the paper, we have used the tool-package PETSc that includes the linear solver “Generalized minimal residual algorithm –GMRES”.

What is completely missing is a discussion on what is the advantage of using the cell method with respect to a consolidated commercial software like COMSOL or research software like GetDP? Flexibility? Speed? Confidence in the results (because no "tricks" have been used inside the software)?

When we solve equation (30) in a certain range of time t, the error accumulated applying this equation is lower than the error accumulated when we use FEM. This is due to the fact that while the thermal mass matrix in FEM has a condition number of 5, the thermal mass matrix that we use in FF-CM has a condition number of 2.7692. This implies that FF-CM is more accurate in the transient thermal analysis than FEM [15].

Besides, as we say in the introduction, in line 42 “In addition, the equations of constitutive type –equations of the medium– are clearly differentiated from the topological type –equations of balance. In FF, the physical laws that govern the electromagnetic equations and the thermal laws of heat transfer associated with magnetic brakes, are expressed in their integral form. In this way, the final system of equations is posed directly, without the need to discretize the equivalent differential equations [11].

The thermal analysis of the magnetic brake with this methodology greatly facilitates the conditions of contour and continuity, when working with global magnitudes and directly raise the system of equations without the need to discretize the differential equations.”

Besides, flexibility is another important factor, because we have access to all what is happening during the calculations, and this increase our confidence in the results.

Please comment why the MLX90614ESF-DCI sensor has been used in addition to a standard thermal camera. Is it because of cost? Or other things like required speed or other variables. If using a commercial thermal camera *only*, probably the whole section III can be shorted a lot. If there are no other reasons, it seems to me that get the temperature through a thermal camera is standard and does not require theory behind it.

The sampling frequency is important in this transitory state study because the disk is rotating. Hence, we need a high sampling frequency. MLX90614 sensor sampling frequency is 100 kHz and FLIR T425 thermal camera sampling frequency is only 9 Hz. So, although we only have information in a single point, we have a high sample frequency for the MLX90614. However, the camera gives information of the whole disk. Besides, accuracy of MLX90614 sensor is higher than FLIR T425 accuracy.

Another reason is that MLX90614 sensor is very cheap, less than fifty euros. However, FLIR T425 thermal camera is very expensive. We have used the collaboration of TBN, S.L Industrial Maintenance Engineering and Integral Lubrication Services, which has lent us the FLIR T425 thermal camera, as you can see in the acknowledgments.

After going further with the paper I did not found a comparison of the two temperature sensors, I wander why. Maybe it would be better to say from the beginning (i.e. in the introduction) that two different methods will be used and carefully explain why two methods to measure the temperature are necessary.

We have not made a direct comparison of the two temperature sensors. We have used two different sensors: an MLX90614 and a camera, to verify the new convective thermal matrix. One takes measures in a point and the other takes measures in matrix array. The measures of both methods have been compared with FEM and CM. But we can compare the visual results obtained for both sensors observing the figures included in the paper.

To enrich and improve the paper, we are going to include in the paper, in the introduction, on line 77, the following paragraph: “We have used two different infrared sensors: a punctual thermal sensor and a camera, to verify the new convective thermal matrix. One takes measures in a point and the other takes measures in matrix array. The measures of both methods have been compared with FEM and CM.

Kryslov should be Krylov.

Thank you. We have corrected this mistake.

I found many information I asked before in the review in the result section. Perhaps it would be desirable to advise the readers in Section 2 that these details will be presented later in Section 4.

Thank you. You are right. We are going to include in the beginning of Section 2, on line 90 the following comment: “The new convective constitutive thermal matrix deduced in this section will be experimentally validated through the experimental data obtained in section 4”.

Reviewer 2 Report

The results of the simulation have been verified by comparing the numerical results with those obtained by the Finite Element Method (FEM) and with experimental data obtained by infrared technology. The difference between the experimental results obtained by infrared sensors and those obtained in the simulations is less than 0.0459%.The paper has been accepted.

Author Response

Thank you very much for your review.

Reviewer 3 Report

The paper presents temperature analysis of magnetic brake by the use of infrared method and 3D cell method with new convective constitutive matrix. Authors investigate thermal field inside conductor disk in transitory regime. How they say, mentioned disk works in motion in magnetic field caused by permanent magnet, what means that current, which induced inside, is able to generate heat. The system is monitored by infrared sensor. Authors propose In addition new thermal convective matrix for the 3D Cell Method in order to simulate and analyze magnetic brake. Obtained results were verified in comparison of numerical results (Finite Element Method) and experimental results. The difference between mentioned methods were very small, less than 0.1%.

The introduction chapter is well organized and written. Authors explain relationship between thermal, electric and magnetic fields. They also indicate the role of infrared methods to analyze and diagnose in industry. The number of used references is correct in my mind.

Chapter 2 – thermal equation of the system – is correct. Authors included all necessary phenomena, which have the impact on the heat value such as thermal conductivity of solid materials of analyzed system.

Chapter 3 – IR method – authors explain the theoretical fundaments, used sensors, IR camera. It is correct but I could not find any information regarding the surface infrared radiation factor, which value is usually less than 1. It is important to know what the value of the factor is to correct measure the temperature. Please complete it indicating the value of the factor for analyzed surface.

Chapter 4 – results – authors present the characteristics of the magnetic brake, numerical validation of CM, experimental validation. I think, the key chapter is well organized and the results are clear presented.

Chapter 5 – conclusions – authors conclude that thanks to their experimentation carried out by IR sensors, CM with formulation of new convective matrix is suitable for simulation temperature aspects for example magnetic brakes. FEM simulations proved the validity of investigated new method. Obtained difference were on small, acceptable level.

Detail comments:

I have seen few grammatical mistake – please review the English.

Formula (1) – I think, all elements of the formula should be explain in the paper text.

The same comments to formula (5).

421 – different used fonts. Please correct.

Author Response

REVIEWER 3

Open Review

English language and style

( ) Extensive editing of English language and style required
( ) Moderate English changes required
(x) English language and style are fine/minor spell check required
( ) I don't feel qualified to judge about the English language and style

As we carefully checked the spelling, we have noted that some paragraphs had the Word corrector in UK English and other in USA English. We have selected the whole document and selected UK English. As a consequence, we have changed a few words in the document as analyse, centre and colour that are highlighted in yellow in the paper on lines 13, 16, 38, 128, 132, 133.

On line 428 we had used a different font for “neodymium” and we have corrected it.

Finally, on line 493 were we said “Kryslov” it should be “Krylov”.

Once we have read in detail the paper we have found two grammatical mistakes. The grammatical construction is the same and follows a subjunctive English structure. So we have changed “has” for “have”, both on lines 63 and 338.

We have also changed Kelvin grades, “°K”, by Kelvin “K” in all the document because Kelvin is not a graduated scale as it is Celsius scale, “°C”.

Yes

Can be   improved

Must be   improved

Not   applicable

Does the   introduction provide sufficient background and include all relevant   references?

(x)

( )

( )

( )

Is the   research design appropriate?

(x)

( )

( )

( )

Are the   methods adequately described?

( )

(x)

( )

( )

Are the   results clearly presented?

(x)

( )

( )

( )

Are the   conclusions supported by the results?

(x)

( )

( )

( )

Comments and Suggestions for Authors

The paper presents temperature analysis of magnetic brake by the use of infrared method and 3D cell method with new convective constitutive matrix. Authors investigate thermal field inside conductor disk in transitory regime. How they say, mentioned disk works in motion in magnetic field caused by permanent magnet, what means that current, which induced inside, is able to generate heat. The system is monitored by infrared sensor. Authors propose in addition new thermal convective matrix for the 3D Cell Method in order to simulate and analyze magnetic brake. Obtained results were verified in comparison of numerical results (Finite Element Method) and experimental results. The difference between mentioned methods were very small, less than 0.1%.

The introduction chapter is well organized and written. Authors explain relationship between thermal, electric and magnetic fields. They also indicate the role of infrared methods to analyze and diagnose in industry. The number of used references is correct in my mind.

Chapter 2 – thermal equation of the system – is correct. Authors included all necessary phenomena, which have the impact on the heat value such as thermal conductivity of solid materials of analyzed system.

Chapter 3 – IR method – authors explain the theoretical fundaments, used sensors, IR camera. It is correct but I could not find any information regarding the surface infrared radiation factor, which value is usually less than 1. It is important to know what the value of the factor is to correct measure the temperature. Please complete it indicating the value of the factor for analyzed surface.

Thank you very much. We forgot to add this information. The surface infrared radiation factor is 0.960. So we have added the following phrase in line 431: “The surface infrared radiation factor is 0.960”.

Chapter 4 – results – authors present the characteristics of the magnetic brake, numerical validation of CM, experimental validation. I think, the key chapter is well organized and the results are clear presented.

Chapter 5 – conclusions – authors conclude that thanks to their experimentation carried out by IR sensors, CM with formulation of new convective matrix is suitable for simulation temperature aspects for example magnetic brakes. FEM simulations proved the validity of investigated new method. Obtained difference were on small, acceptable level.

Detail comments:

I have seen few grammatical mistake – please review the English.

As we have previously said, we have carefully reviewed the English and we have found a few mistakes that have been corrected.

Formula (1) – I think, all elements of the formula should be explained in the paper text.

Thank you for your comments. The explanation of variables a and f  have been added on line 108 in the text. “Where the degrees of freedom a and f are the magnetic potential and electric potential, respectively”. We have also added a in the nomenclature Table.

The same comments to formula (5).

Thank you for your corrections. The explanation of variables MrCp and Ml have been added to the text on line 143: “,  is the constitutive matrix in heat transmission in transitory state and  is the thermal conductivity constitutive matrix.

421 – different used fonts. Please correct.

Thank you. We have changed the fonts.

Round  2

Reviewer 1 Report

Thank you for having improved the paper. I judge the paper mature for publication.

Reviewer 3 Report

The paper presents temperature analysis of magnetic brake by the use of infrared method and 3D cell method with new convective constitutive matrix. Authors investigate thermal field inside conductor disk in transitory regime. How they say, mentioned disk works in motion in magnetic field caused by permanent magnet, what means that current, which induced inside, is able to generate heat. The system is monitored by infrared sensor. Authors propose In addition new thermal convective matrix for the 3D Cell Method in order to simulate and analyze magnetic brake. Obtained results were verified in comparison of numerical results (Finite Element Method) and experimental results. The difference between mentioned methods were very small, less than 0.1%.

Authors included all my sugestions. they completed information regarding surface infrared radiation factor, which value is usually less than 1.

They corrected grammatical mistake, I have noticed.

they described all elements of formula 1 and 5 in the paper text, I asked in my comments.

Authors corrected the font size in paper text, I asked.

I think, the paper is ready to be published in present form.